# A Hierarchy of Probability, Fluid and Generalized Densities for the Eulerian Velocivolumetric Description of Fluid Flow, for New Families of Conservation Laws

**DOI:** 10.3390/e24101493

**Published:** 2022-10-19

**Authors:** Robert K. Niven

**Affiliations:** School of Engineering and Information Technology, The University of New South Wales, Canberra, ACT 2600, Australia; r.niven@adfa.edu.au

**Keywords:** Reynolds transport theorem, conservation laws, Lie derivative, differential geometry, probability density function, Eulerian description, fluid densities

## Abstract

The Reynolds transport theorem occupies a central place in continuum mechanics, providing a generalized integral conservation equation for the transport of any conserved quantity within a fluid or material volume, which can be connected to its corresponding differential equation. Recently, a more generalized framework was presented for this theorem, enabling parametric transformations between positions on a manifold or in any generalized coordinate space, exploiting the underlying continuous multivariate (Lie) symmetries of a vector or tensor field associated with a conserved quantity. We explore the implications of this framework for fluid flow systems, based on an Eulerian velocivolumetric (position-velocity) description of fluid flow. The analysis invokes a hierarchy of five probability density functions, which by convolution are used to define five fluid densities and generalized densities relevant to this description. We derive 11 formulations of the generalized Reynolds transport theorem for different choices of the coordinate space, parameter space and density, only the first of which is commonly known. These are used to generate a table of integral and differential conservation laws applicable to each formulation, for eight important conserved quantities (fluid mass, species mass, linear momentum, angular momentum, energy, charge, entropy and probability). The findings substantially expand the set of conservation laws for the analysis of fluid flow and dynamical systems.

## 1. Introduction

Near the end of his distinguished career, Osborne Reynolds presented what is now called the *Reynolds transport theorem*: a generalized conservation equation for the transport of a conserved quantity within a body of fluid (the *domain*, *fluid volume* or *material volume*) as it moves through a prescribed region of space (the *control volume*) [1]. This provides a universal formulation for the construction of integral conservation equations for any conserved quantity, and can be used to derive the corresponding differential equations for these quantities, e.g., [2,3,4,5,6]. Such conservation laws—founded on the paradigm of the field or Eulerian description of fluid flow—provide the basis for most theoretical and numerical analyses of flow systems. Extensions of the Reynolds transport theorem have been presented for moving and smoothly-deforming control volumes [3,6,7], domains with fixed or moving discontinuities [7,8], irregular and rough domains [9,10], two-dimensional domains [11,12,13,14], and differentiable manifolds using the formalism of exterior calculus [15,16,17,18,19]. The Reynolds transport theorem is also a special case of the Helmholtz transport theorem for flow through an open and moving surface [20], and of stochastic formulations to incorporate random diffusion and uncertainty [21,22].

Traditionally, the Reynolds transport theorem has been viewed exclusively as a continuous one-parameter (temporal) mapping of the density of a conserved quantity in geometric space, along the pathlines described by a time-dependent velocity vector field. Indeed, the above formulations all conform to this tradition. However, Flanders [15] interpreted the theorem more broadly as a generalization of the Leibniz rule for differentiation of an integral, rather than simply a construct of continuum mechanics. It is therefore far more general and powerful than the traditional interpretation might suggest. Using this insight, a generalized framework for the Reynolds transport theorem has recently been presented [23,24], based on continuous multiparametric mappings of a differential form on a manifold—or of a density within a generalized coordinate space—connected by the maximal integral curves or surfaces described by a vector or tensor field. This extends the traditional interpretation to encompass new *transformation theorems* in any parameter space, not just in time; these exploit previously unreported multiparametric continuous (Lie) symmetries associated with a conserved quantity in the space considered. These can be used, for example, to connect different positions in a velocity space connected by a velocity gradient tensor field, different positions in a Fourier spectral space connected by a velocity-wavenumber tensor field, or different positions in a velocity and chemical species space connected by velocity and concentration gradient tensor fields, for the analysis of chemical reaction systems and plasmas [23]. The generalized framework also yields new forms of the Liouville equation for the conservation of probability in different spaces, and of the Perron-Frobenius and Koopman operators for the extrapolation of probability densities or observable densities in such systems [23].

The aim of this work is to examine the implications of the generalized Reynolds transport theorem—and consequential integral and differential conservation laws—for an Eulerian velocivolumetric (position-velocity) description of fluid flow systems. The analysis commences in Section 2 with a detailed discussion of the extended Eulerian description, and of the properties of the volumetric and velocimetric domains for several well-known classes of fluid flow systems. This leads in Section 3 to a hierarchy of densities, starting in Section 3.1 with five probability density functions (pdfs), which are defined and in which their commutative relations are examined. These are used in Section 3.2 to define five corresponding fluid or material densities, of which only the volumetric density ρ is commonly used in continuum mechanics; the remaining four have many similarities to other densities (such as the phase space density) used in other branches of physics. The fluid densities are formally defined from the pdfs by mathematical convolutions, for which the definitions and philosophical implications are discussed in Appendices Appendix A and Appendix B. The fluid densities are then used in Section 3.3 to define corresponding generalized densities for any conserved quantity. In Section 4, we present the generalized framework for the Reynolds transport theorem in both exterior calculus and vector calculus formulations. As explained in Section 4 and Appendix C, this can also be used to extract Lie or partial differential equations for an individual fluid element, the former expressed in terms of the Lie derivative of a volume form in the domain. In Section 5, these equations are then used to generate 11 formulations of the Reynolds transport theorem arising from the velocivolumetric description, for different choices of the coordinate space, parameter space and density. Of these, only the first accords with the traditional Reynolds transport theorem [1]. For each formulation, a table of integral and differential conservation equations is presented for the eight conserved quantities commonly considered in fluid mechanics (fluid mass, species mass, linear momentum, angular momentum, energy, charge, entropy and probability). The analyses provide a considerable assortment of new conservation laws for the analysis of fluid flow systems.

In the following sections, the mathematical notation is defined when first used and is also listed in Table 1.

## 2. The Position-Velocity Description and Domains

In continuum mechanics, fluid flow systems are commonly examined using the *Eulerian description*, in which each local property of the fluid is specified as a function of position (such as in Cartesian coordinates) $x={\left[x,y,z\right]}^{⊤}\in \Omega \subset R^{3}$ and time t∈R as the fluid moves past, where Ω is a three-dimensional geometric space and ${}^{⊤}$ is the transpose. Thus, for example, fluid mechanicians commonly consider the three-dimensional velocity u(x,t), the volumetric mass density ρ(x,t) and the volumetric mass concentration $\rho_{c}\left(x,t\right)$ of the *c*th chemical species within the Eulerian description. We here consider an extended *velocivolumetric continuum description* of a dynamical system based on Eulerian velocity and position coordinates, in which each local property of a fluid is specified as a function of the instantaneous fluid velocity $u={\left[u,v,w\right]}^{⊤}\in D\subset R^{3}$, position $x\in \Omega \subset R^{3}$ and time t∈R as the fluid moves past, where D is a three-dimensional velocity space. This treatment—somewhat analogous to the phase space description used in many branches of physics—has the advantage of explicitly incorporating the velocity dependence of physical quantities, significantly extending the breadth of physical quantities that can be considered, and the scope and fidelity of the analyses.

We consider two alternative representations of the paired domains Ω and D: (a)The *geometric representation*—the usual physical viewpoint—in which D(x,t) is a function of position and time, and Ω(t) is a function of time. In this perspective, as illustrated in Figure 1a, there exists a map between each position x∈Ω(t) and an entire velocity space D(x,t), consisting of all possible velocities for this position and time.(b)The *velocimetric representation*—an alternative viewpoint—in which Ω(u,t) is a function of velocity and time, and D(t) is a function of time. In this perspective, as illustrated in Figure 1b, there exists a map between each velocity u∈D(t) and an entire geometric space Ω(u,t), consisting of all possible positions for this velocity and time.

In principle, the set of all ordered triples (u,x,t) for a given flow system can be mapped into either of these representations, hence D(x,t)⊆D(t) for all x∈Ω(t) and Ω(u,t)⊆Ω(t) for all u∈D(t).

Examining the velocity domain D(t) or D(x,t), we make two important assertions. First, we expect D(t) or D(x,t) to be continuous, since for most flows, it is physically impossible for a local velocity to change from $u_{1}$ to $u_{2}$ without passing through all intermediate velocities $u_{1}<u<u_{2}$, however fleetingly. The main exception to this rule are flows that cross a regime threshold, e.g., from laminar to turbulent flow, or subsonic to supersonic flow, leading to two internally continuous velocity domains $D_{1}$ and $D_{2}$, which may be disjoint. Second, the velocity cannot be infinite (positive or negative) for any physically realizable flow, as this would require local velocities of infinite kinetic energy. In consequence, for all flows the domains D(t) or D(x,t) should be bounded, and for many flows will also be closed, and thence compact. We note in passing that integration of the velocity over $R^{3}$ is a useful technique for many calculations, but this invokes an approximation that cannot be manifested physically. In its place, integration over a compact velocity domain D(t) or D(x,t) is well-defined, while integration over bounded and open domains can be performed by careful consideration of the limits or, if necessary, by transformation to the Lebesgue integral.

In consequence, each domain in Figure 1 is drawn as compact and simply connected, a useful starting assumption for turbulent flow systems, but with many exceptions. For most flows involving a continuous fluid volume with no velocity discontinuities, the primary domains Ω(t) and D(t) and the subsidiary domain D(x,t) should be compact and simply connected, but some subsidiary geometric spaces Ω(u,t) may consist of disjoint subspaces, each associated with a different location (or set of locations) within the flow. Consider, for example, two-dimensional turbulent Poiseuille flow between parallel plates: each pair of positions $y_{+}$ and $y_{-}$ symmetric about the centerline will have the same mean velocity, hence many instantaneous velocities u∈D(t) will map to two disjoint geometric subspaces $\Omega_{+}\left(u,t\right)$ and $\Omega_{-}\left(u,t\right)$, respectively, containing $y_{+}$ and $y_{-}$. By breaking the original problem into smaller coupled flows, or by a judicious choice of coordinate system, it should be possible to isolate or unite these subspaces. In flows with different flow regimes, leading to disjoint velocity domains D(t) and D(x,t), it should be possible to isolate each subdomain using a dimensionless discriminator (such as a Reynolds, Mach or Froude number). For some flows, for example the turbulent boundary layer, there are long-standing arguments over whether the overall geometric space Ω(t) can be considered compact, due to a lack of boundedness or closedness, but despite this the subsidiary geometric spaces Ω(u,t) and the overall and subsidiary velocity domains D(t) and D(x,t) will very likely be bounded and may also be closed. For homogeneous isotropic turbulence—a highly idealized flow—each subsidiary domain D(x,t) or Ω(u,t) must be independent, respectively, of position or velocity, and so the two representations will collapse to give two separable domains Ω(t) and D(t). In laminar flows, each subsidiary velocity domain D(x,t) can be idealized as a single point (in reality, allowing for fluctuations, a small region), while each subsidiary geometric space Ω(u,t) will consist of single or multiple disconnected points (or small regions).

## 3. A Hierarchy of Densities

### 3.1. Probability Density Functions

We can now define the primary probability density functions (pdfs) that underlie continuum systems, and will subsequently be used to define physical densities in these systems. Writing the nonnegative real line as $R_{0}^{+}$, the velocivolumetric description gives rise to the following five pdfs:(a)A volumetric pdf $p\left(x|t\right):\Omega \times R\to R_{0}^{+}$ [SI units: m^−3^];(b)A velocimetric pdf $p\left(u|t\right):D\times R\to R_{0}^{+}$ [(m s^−1^)^−3^];(c)A velocivolumetric pdf $p\left(u,x|t\right):D\times \Omega \times R\to R_{0}^{+}$ [m^−3^ (m s^−1^)^−3^];(d)A conditional velocimetric (ensemble) pdf $p\left(u|x,t\right):D\times \Omega \times R\to R_{0}^{+}$ [(m s^−1^)^−3^]; and(e)A conditional volumetric pdf $p\left(x|u,t\right):\Omega \times D\times R\to R_{0}^{+}$ [m^−3^];

where the solidus | is the conditional probability symbol, with the conditions listed to the right (these follow the standard pdf notation with a common symbol *p*, leading to a mixed signature and functional notation). Generally, the pdf at each point forms part of a probability density field defined throughout its domain.

Using the notation $dV=dxdydz=d^{3}x$ for an infinitesimal volume element and $dU=dudvdw=d^{3}u$ for an infinitesimal three-dimensional velocity element, the five pdfs will by definition satisfy the following nine relations: (1)$$\begin{array}{cc} 1 & ={\;\int \int \int \;}_{\Omega (t)}p\left(x|t\right)\;dV \end{array}$$
(2)$$\begin{array}{cc} 1 & ={\;\int \int \int \;}_{D(t)}p\left(u|t\right)\;dU \end{array}$$
(3)$$\begin{array}{c} \begin{array}{cc} 1 & ={\;\int \int \int \;}_{\Omega (t)}{\;\int \int \int \;}_{D(x,t)}p\left(u,x|t\right)\;dU\;dV={\;\int \int \int \;}_{D(t)}{\;\int \int \int \;}_{\Omega (u,t)}p\left(u,x|t\right)\;dV\;dU \end{array} \end{array}$$
(4)$$\begin{array}{cc} p(x|t) & ={\;\int \int \int \;}_{D(x,t)}p\left(u,x|t\right)\;dU \end{array}$$
(5)$$\begin{array}{cc} p(u|t) & ={\;\int \int \int \;}_{\Omega (u,t)}p\left(u,x|t\right)\;dV \end{array}$$
(6)$$\begin{array}{cc} p(u|x,t) & =\frac{p(u,x|t)}{p(x|t)}=\frac{p(u,x|t)}{{\;\int \int \int \;}_{D(x,t)}p\left(u,x|t\right)dU} \end{array}$$
(7)$$\begin{array}{cc} p(x|u,t) & =\frac{p(u,x|t)}{p(u|t)}=\frac{p(u,x|t)}{{\;\int \int \int \;}_{\Omega (u,t)}p\left(u,x|t\right)dV} \end{array}$$
(8)$$\begin{array}{cc} 1 & ={\;\int \int \int \;}_{D(x,t)}p\left(u|x,t\right)\;dU \end{array}$$
(9)$$\begin{array}{cc} 1 & ={\;\int \int \int \;}_{\Omega (u,t)}p\left(x|u,t\right)\;dV \end{array}$$
The connections between the five pdfs, and the roles of the different domain representations, are illustrated in the relational diagram in Figure 2. As evident, the geometric representation is adopted for the integration paths on the left-hand side of Figure 2, while the velocimetric representation is required on the right-hand side.

Designating the infinitesimal velocity element, position element and time interval, respectively, by [u,u+du], [x,x+dx] and [t,t+dt], we examine each pdf in turn:(a)The velocivolumetric pdf p(u,x|t) is the most fundamental of the pdfs, giving rise to p(x|t) or p(u|t) by the marginalization operations in Equations (Equation 4) and (Equation 5), and p(u|x,t) and p(x|u,t) by the conditioning operations in Equations (Equation 6) and (Equation 7). Physically—albeit imprecisely [25,26]—we can interpret p(u,x|t)dUdV as the joint probability of an infinitesimal fluid element having a velocity of [u,u+du] and position in [x,x+dx], during the time interval [t,t+dt].(b)The volumetric pdf p(x|t) can be recognized as the common probabilistic descriptor for fluid flow systems, forming the basis of the fluid mechanics formulations of the Liouville and Fokker–Planck Equations [27,28,29,30,31], and allied to the volumetric density ρ(x,t). Physically, p(x|t)dV can be interpreted as the probability that a fluid element is situated at the position [x,x+dx] in the time interval [t,t+dt], regardless of velocity.(c)The velocimetric pdf p(u|t) is rather strange. Physically, p(u|t)dU can be interpreted as the probability of fluid elements within the control volume having a velocity of [u,u+du] in the time interval [t,t+dt], regardless of position.(d)To understand the conditional velocimetric pdf p(u|x,t), we interpret p(u|x,t)dU as the probability that a fluid element has a velocity of [u,u+du], at the position [x,x+dx] and time [t,t+dt]. We therefore recognize p(u|x,t)—typically but incorrectly written as p(u)—as the basis of the ensemble mean commonly used in the Reynolds-averaged Navier–Stokes (RANS) equations, and of the single-position correlation functions of turbulent fluid mechanics [32,33,34,35,36].(e)To understand the conditional volumetric pdf p(x|u,t), we interpret p(x|u,t)dV as the probability that a fluid element has a position in [x,x+dx], for a velocity of [u,u+du] and time [t,t+dt].
The infinitesimal intervals are necessitated by the fact that the probability itself vanishes at each point: for example, for the random variable *X* with values *x*, by definition Prob(X=x)
$=\lim_{dx\to 0}\mathrm{Prob}\left(x\leq X<x+dx\right)=\lim_{dx\to 0}\int_{x}^{x+dx}p\left(x\right)dx=\int_{x}^{x}p\left(x\right)dx=0$, e.g., [25,37].

In addition to the pdfs in Figure 2, it is possible to consider joint pdfs with respect to time, including p(x,t), p(u,t) and p(u,x,t). These require normalization over a time interval in addition to their volume and/or velocity domain(s). Such pdfs are closely associated with path-based formulations for the description of entire histories of events, e.g., [38,39,40], and are not considered further here.

### 3.2. Fluid or Material Densities

The above five pdfs can be used to define corresponding fluid or material mass densities, four of which are not commonly used for the analysis of continuum systems (these revert to standard signature and functional notation):(a)A volumetric fluid density $\rho :\Omega \times R\to R_{0}^{+}$, (x,t)↦ρ(x,t) [kg m^−3^];(b)A velocimetric fluid density Д: $D\times R\to R_{0}^{+}$, (u,t)↦(u,t) [kg (m s^−1^)^−3^];(c)A velocivolumetric fluid density $\zeta :D\times \Omega \times R\to R_{0}^{+}$, (u,x,t)↦ζ(u,x,t) [kg m^−3^ (m s^−1^)^−3^];(d)A conditional velocimetric (ensemble) fluid density $\eta :D\times \Omega \times R\to R_{0}^{+}$, (u,x,t)↦η(u,x,t) [kg (m s^−1^)^−3^]; and(e)A conditional volumetric fluid density $\xi :D\times \Omega \times R\to R_{0}^{+}$, (u,x,t)↦ξ(u,x,t) [kg m^−3^];

As with the pdfs, each density forms part of a density field, defined throughout its domain. The symbols for the last four densities, including the Cyrillic “de” character (from the transliteration of “density”), are chosen to not conflict with the most common notation of fluid mechanics.

For analysis, consider a material volume containing the fluid or material mass *M*, which in the absence of sources or sinks of fluid will be constant in time (n.b., systems with sources or sinks of fluid mass will require a more complicated treatment). First, we require the five fluid densities to satisfy the following nine relations:(10)$$\begin{array}{cc} M & ={\;\int \int \int \;}_{\Omega (t)}\rho \left(x,t\right)\;dV \end{array}$$
(11)$$\begin{array}{cc} M & ={\;\int \int \int \;}_{D(t)}Д\left(u,t\right)\;dU \end{array}$$
(12)$$\begin{array}{c} \begin{array}{cc} M & ={\;\int \int \int \;}_{\Omega (t)}{\;\int \int \int \;}_{D(x,t)}\zeta \left(u,x,t\right)\;dU\;dV={\;\int \int \int \;}_{D(t)}{\;\int \int \int \;}_{\Omega (u,t)}\zeta \left(u,x,t\right)\;dV\;dU \end{array} \end{array}$$
(13)$$\begin{array}{cc} \rho (x,t) & ={\;\int \int \int \;}_{D(x,t)}\zeta \left(u,x,t\right)\;dU \end{array}$$
(14)$$\begin{array}{cc} Д(u,t) & ={\;\int \int \int \;}_{\Omega (u,t)}\zeta \left(u,x,t\right)\;dV \end{array}$$
(15)$$\begin{array}{cc} \eta (u,x,t) & =\frac{\zeta (u,x,t)\;M}{\rho (x,t)}=\frac{\zeta (u,x,t)}{\left(\frac{{\;\int \int \int \;}_{D(x,t)}\zeta \left(u,x,t\right)dU}{M}\right)}=\frac{\zeta (u,x,t)}{\left(\frac{{\;\int \int \int \;}_{D(x,t)}\zeta \left(u,x,t\right)dU}{{\;\int \int \int \;}_{\Omega (t)}{\;\int \int \int \;}_{D(x,t)}\zeta \left(u,x,t\right)\;dU\;dV}\right)} \end{array}$$
(16)$$\begin{array}{cc} \xi (u,x,t) & =\frac{\zeta (u,x,t)\;M}{Д(u,t)}=\frac{\zeta (u,x,t)}{\left(\frac{{\;\int \int \int \;}_{\Omega (u,t)}\zeta \left(u,x,t\right)dV}{M}\right)}=\frac{\zeta (u,x,t)}{\left(\frac{{\;\int \int \int \;}_{\Omega (u,t)}\zeta \left(u,x,t\right)dV}{{\;\int \int \int \;}_{D(t)}{\;\int \int \int \;}_{\Omega (u,t)}\zeta \left(u,x,t\right)\;dV\;dU}\right)} \end{array}$$
(17)$$\begin{array}{cc} M & ={\;\int \int \int \;}_{D(x,t)}\eta \left(u,x,t\right)\;dU \end{array}$$
(18)$$\begin{array}{cc} M & ={\;\int \int \int \;}_{\Omega (u,t)}\xi \left(u,x,t\right)\;dV \end{array}$$
Clearly these are analogs of Equations (Equation 1)–(Equation 9), modified only by the introduction of *M* into the definitions of η and ξ. Note that the time dependence is lost in the integrations in Equations (Equation 10)–(Equation 12) and Equations (Equation 17) and (Equation 18); furthermore Equations Equation 17) and (Equation 18) also lose their dependence, respectively, on x or u. The connections between the different fluid densities are illustrated in the relational diagram in Figure 3.

Second, following a rich line of research, e.g., [41,42,43,44,45,46,47,48,49,50,51,52,53,54,55,56,57,58,59,60,61,62,63,64,65,66,67,68], the fluid or material density ρ at each position x and time *t* can be rigorously defined by integration over a small fluid volume V—or equivalently a small fluid mass m—for which there are two interpretations. In the common viewpoint, V must be sufficiently large to enable the fluid to be considered a continuum, e.g., [44,45,46,47,56,57]. Thus, in single phase systems, it must provide a “microscopic” scale large enough to average out the molecular phenomena, while for multiphase systems it may need to be larger than the dominant “macroscopic” scales [45,48,58]. However, in the contrary “relativist” viewpoint, V is not considered a property of the continuum, but is simply a characteristic of the measurement scale [53,54,59,61]. The analyses here are agnostic, encompassing both viewpoints; we also adopt a small velocimetric domain U for velocity averaging. For each small integral, we adopt the local Cartesian position coordinates $r={\left[r_{x},r_{y},r_{z}\right]}^{⊤}\in \Omega \subset R^{3}$ and velocity coordinates $s={\left[s_{u},s_{v},s_{w}\right]}^{⊤}\in D\subset R^{3}$ aligned with their corresponding global coordinates, and with their origin, respectively, at x or u. The small domains V and U are functions, respectively, of x and u, as well as time, and must conform to the two domain representations introduced earlier, bringing additional dependencies on r and/or s into the definition of ζ. The five fluid or material densities can then be defined from their underlying pdfs by the following convolutions: (19)$$\begin{array}{c} \begin{array}{cc} {}[\rho ](x,t)\; & =\int_{m(x,t)}p\left(r|t\right)\;dm\left(x+r,t\right)={\;\int \int \int \;}_{V(x,t)}p\left(r|t\right)\;\rho \left(x+r,t\right)\;dV\left(r,t\right) \end{array} \end{array}$$
(20)$$\begin{array}{c} \begin{array}{cc} \langle Д\rangle (u,t)\; & =\int_{m(u,t)}p\left(s|t\right)\;dm\left(u+s,t\right)={\;\int \int \int \;}_{U(u,t)}p\left(s|t\right)\;\left(u+s,t\right)\;dU\left(s,t\right) \end{array} \end{array}$$
(21)$$\begin{array}{cc} \left[\langle \zeta \rangle ](u,x,t\right) & =\int_{m(u,x,t)}p\left(s,r|t\right)dm\left(u+s,x+r,t\right)\\ \; & =\underset{V(x,t)}{\;\int \int \int \;}\underset{U(u,x+r,t)}{\;\int \int \int \;}p\left(s,r|t\right)\zeta \left(u+s,x+r,t\right)dU\left(s,x+r,t\right)dV\left(s,r,t\right)\\ \; & =\underset{U(u,t)}{\;\int \int \int \;}\underset{V(u+s,x,t)}{\;\int \int \int \;}p\left(s,r|t\right)\zeta \left(u+s,x+r,t\right)dV\left(u+s,r,t\right)dU\left(s,r,t\right) \end{array}$$
(22)$$\begin{array}{c} \begin{array}{cc} \langle \eta \rangle (u,x,t)\; & =\int_{m(u,x,t)}p\left(s|x,t\right)\;dm\left(u+s,x,t\right)={\;\int \int \int \;}_{U(u,x,t)}p\left(s|x,t\right)\;\eta \left(u+s,x,t\right)\;dU\left(s,x,t\right) \end{array} \end{array}$$
(23)$$\begin{array}{c} \begin{array}{cc} {}[\xi ](u,x,t)\; & =\int_{m(u,x,t)}p\left(r|u,t\right)\;dm\left(u,x+r,t\right)={\;\int \int \int \;}_{V(u,x,t)}p\left(r|u,t\right)\;\xi \left(u,x+r,t\right)\;dV\left(u,r,t\right) \end{array} \end{array}$$
where [·] is a local volumetric expectation, 〈·〉 is a local velocimetric expectation, and dm is an infinitesimal element of fluid mass. A more detailed discussion of these definitions is given in Appendix A, while the philosophical implications of the use of pdfs to define material densities are explored in Appendix B. Note that if each pdf is assumed uniformly distributed over its domain, each expected fluid density reduces to the product of its underlying pdf and the fluid mass, as would be obtained from dimensional considerations.

In this study, we adopt the mass integrals in Equations (Equation 19)–(Equation 23) as the primary definitions of the fluid densities, since the volumetric and velocimetric integrals require knowledge of point density terms that need to be defined. For convenience the expectation notations used in Equations (Equation 19)–(Equation 23) are now dropped.

How should we physically interpret the different fluid or material densities? Several schematic diagrams to aid their interpretation are given in Figure 4. From their functional forms and underlying pdfs, we see that ξ, η, ζ and ρ are local densities, i.e., they apply to each infinitesimal volume element within the geometric space. In contrast, Д is a non-local density, applying over the entire geometric space Ω. Furthermore, ξ, η, ζ and Д also apply to infinitesimal velocity elements within the velocity domain. Considering each density in turn:(a)As shown in Figure 4a, the velocivolumetric density ζ represents the fluid mass per unit of velocimetric and geometric space carried by an infinitesimal fluid element of velocity [u,u+du] through the infinitesimal control volume element at [x,x+dx], during the infinitesimal time interval [t,t+dt]. In consequence, ζ is both a velocity spectral density and a local volumetric density, accounting for the distribution of fluid mass with both velocity and position. As evident in Figure 3, ζ is central to the current formulation, giving the other fluid densities by marginalization or conditioning.(b)As shown in Figure 4b, the well-known volumetric fluid density ρ represents the fluid mass per unit volume carried by the fluid through the infinitesimal control volume element at [x,x+dx], during the time interval [t,t+dt]. From Equation (Equation 13), ρ is obtained by integration (marginalization) of ζ over the subsidiary velocity domain D(x,t), consisting of all realizable velocities for this position and time. In well-behaved systems, D(x,t) should consist of an infinitesimal trajectory (or trajectory bundle) in velocity space, from which it may be possible to calculate ρ by line integration.(c)In contrast, as shown in Figure 4c, the velocimetric density Д represents the fluid mass per unit of velocimetric space transported by fluid elements of velocity [u,u+du] throughout the control volume, during the time interval [t,t+dt]. This is a very strange, aggregated density field, representing the distribution of fluid mass across the velocity spectrum rather than with position, but nonetheless both it and its underlying pdf p(u|t) are well-defined. From Equation (Equation 14), Д is obtained by integration (marginalization) of ζ over the subsidiary geometric space Ω(u,t), consisting of all realizable positions for this velocity and time. As discussed in Section 2 and illustrated in Figure 4c, in many flow systems Ω(u,t) will consist of several disjoint but continuous domains, which depending on the flow system may be bounded and may also be closed.(d)The conditional ensemble density η (not illustrated) represents the fluid mass per unit velocimetric space carried by a fluid element of velocity [u,u+du], at the position [x,x+dx] during the time interval [t,t+dt]. From Equation (Equation 15), η is obtained by the ratio of ζM and ρ, which can be interpreted as a conditioning operation over position. This removes the volume from the dimensions of η, giving the units of fluid mass per unit velocity space.(e)The conditional density ξ (not illustrated) represents the fluid mass per unit volume carried by a fluid element in the position [x,x+dx], of velocity [u,u+du] during the time interval [t,t+dt]. From Equation (Equation 16), ξ is obtained by the ratio of ζM and Д, which can be interpreted as a conditioning operation over velocity. This removes the velocity volume from the dimensions of ξ, giving the units of fluid mass per unit volume.

The distinctions between ξ, η and ζ are therefore quite subtle, but since they arise from separate underlying pdfs, each density is mathematically well-defined. We further see that Figure 3 is an analog of Figure 2, exhibiting the same vertical symmetry, with the geometric representation evident on the left-hand side (integration paths in Equations (Equation 10), (Equation 13) and (Equation 17)), and the velocimetric representation evident on the right-hand side (integration paths in Equations (Equation 11), (Equation 14) and (Equation 18)).

### 3.3. Generalized Densities

Based on the above definitions, we can now construct five generalized densities of any conserved quantity carried by a fluid flow (strictly, density fields of an extensive variable):(a)Volumetric densities $\alpha :\Omega \times R\to R_{0}^{+}$, (x,t)↦α(x,t) [qty m^−3^];(b)Velocimetric densities $\beta :D\times R\to R_{0}^{+}$, (u,t)↦β(u,t) [qty (m s^−1^)^−3^];(c)Velocivolumetric densities $\varphi :D\times \Omega \times R\to R_{0}^{+}$, (u,x,t)↦φ(u,x,t) [qty m^−3^ (m s^−1^)^−3^];(d)Conditional velocimetric (ensemble) densities $\theta :D\times \Omega \times R\to R_{0}^{+}$, (u,x,t)↦θ(u,x,t) [qty (m s^−1^)^−3^]; and(e)Conditional volumetric densities $\epsilon :D\times \Omega \times R\to R_{0}^{+}$, (u,x,t)↦ϵ(u,x,t) [qty m^−3^];

where “qty” denotes the units of the conserved quantity. These can be defined by the following relations:(24)$$\begin{array}{cc} \alpha (x,t) & =\rho \left(x,t\right)\;\underline{\alpha }\left(x,t\right) \end{array}$$
(25)$$\begin{array}{cc} \beta (u,t) & =Д\left(u,t\right)\;\overset{˘}{\beta }\left(u,t\right) \end{array}$$
(26)$$\begin{array}{cc} \varphi (u,x,t) & =\zeta \left(u,x,t\right)\;\underline{\overset{˘}{\varphi }}\left(u,x,t\right) \end{array}$$
(27)$$\begin{array}{cc} \theta (u,x,t) & =\eta \left(u,x,t\right)\;\underline{\overset{˘}{\theta }}\left(u,x,t\right) \end{array}$$
(28)$$\begin{array}{cc} \epsilon (u,x,t) & =\xi \left(u,x,t\right)\;\underline{\overset{˘}{\epsilon }}\left(u,x,t\right) \end{array}$$
where $\underline{\alpha }$, $\overset{˘}{\beta }$, $\underline{\overset{˘}{\varphi }}$, $\underline{\overset{˘}{\theta }}$ and $\underline{\overset{˘}{\epsilon }}$ [qty kg^−1^] are specific quantities, representing the quantity carried per unit fluid or material mass. For precision, these are labeled by an underline or a breve accent to designate their functional dependencies; e.g., for the specific energy, $\underline{e}$ indicates a local density (a function of x), $\overset{˘}{e}$ indicates a velocity-distinct density (a function of u), and $\underline{\overset{˘}{e}}$ indicates dependence on both u and x (hence $\underline{e}={\;\int \int \int \;}_{D(x,t)}\underline{\overset{˘}{e}}\;dU$ and $\overset{˘}{e}={\;\int \int \int \;}_{\Omega (t)}\underline{\overset{˘}{e}}\;dV$). However, the specific momentum density (the local velocity) u does not require these designations—being already velocity-dependent—provided that care is taken over its dependence on position.

We also define Q(t) to be the total conserved quantity (of any type) in the volumetric domain, which due to sources or sinks of the conserved quantity will in general be a function of time *t*. The generalized densities will then satisfy the following nine integral relations: (29)$$\begin{array}{cc} Q(t) & ={\;\int \int \int \;}_{\Omega (t)}\alpha \left(x,t\right)\;dV \end{array}$$
(30)$$\begin{array}{cc} Q(t) & ={\;\int \int \int \;}_{D(t)}\beta \left(u,t\right)\;dU \end{array}$$
(31)$$\begin{array}{c} \begin{array}{cc} Q(t) & ={\;\int \int \int \;}_{\Omega (t)}{\;\int \int \int \;}_{D(x,t)}\varphi \left(u,x,t\right)\;dU\;dV={\;\int \int \int \;}_{D(t)}{\;\int \int \int \;}_{\Omega (u,t)}\varphi \left(u,x,t\right)\;dV\;dU \end{array} \end{array}$$
(32)$$\begin{array}{cc} \alpha (x,t) & ={\;\int \int \int \;}_{D(x,t)}\varphi \left(u,x,t\right)\;dU \end{array}$$
(33)$$\begin{array}{cc} \beta (u,t) & ={\;\int \int \int \;}_{\Omega (u,t)}\varphi \left(u,x,t\right)\;dV \end{array}$$
(34)$$\begin{array}{cc} \theta (u,x,t) & =\frac{\varphi (u,x,t)\;Q(t)}{\alpha (x,t)}=\frac{\varphi (u,x,t)}{\left(\frac{{\;\int \int \int \;}_{D(x,t)}\varphi \left(u,x,t\right)dU}{Q(t)}\right)}=\frac{\varphi (u,x,t)}{\left(\frac{{\;\int \int \int \;}_{D(x,t)}\varphi \left(u,x,t\right)dU}{{\;\int \int \int \;}_{\Omega (t)}{\;\int \int \int \;}_{D(x,t)}\varphi \left(u,x,t\right)\;dU\;dV}\right)} \end{array}$$
(35)$$\begin{array}{cc} \epsilon (u,x,t) & =\frac{\varphi (u,x,t)\;Q(t)}{\beta (u,t)}=\frac{\varphi (u,x,t)}{\left(\frac{{\;\int \int \int \;}_{\Omega (u,t)}\varphi \left(u,x,t\right)dV}{Q(t)}\right)}=\frac{\varphi (u,x,t)}{\left(\frac{{\;\int \int \int \;}_{\Omega (u,t)}\varphi \left(u,x,t\right)dV}{{\;\int \int \int \;}_{D(t)}{\;\int \int \int \;}_{\Omega (u,t)}\varphi \left(u,x,t\right)\;dV\;dU}\right)} \end{array}$$
(36)$$\begin{array}{cc} Q(t) & ={\;\int \int \int \;}_{D(x,t)}\theta \left(u,x,t\right)\;dU \end{array}$$
(37)$$\begin{array}{cc} Q(t) & ={\;\int \int \int \;}_{\Omega (u,t)}\epsilon \left(u,x,t\right)\;dV \end{array}$$
Clearly these are analogs of Equations (Equation 10)–(Equation 18), containing Q(t) rather than *M*. The connections between generalized densities are shown in the relational diagram in Figure 5 (compare Figure 2 and Figure 3). By synthesis of Equations (Equation 19)–(Equation 23) and (Equation 24)–(Equation 28), we can also extract the relations between each generalized density and its underlying pdf (see discussion in Appendix A).

Finally, we note that all formulations in this section are functions of time, and in general of both position and velocity. These can be simplified to give quantities and domains that are not functions of time, position or velocity, leading to a considerable assortment of reduced mathematical formulations.

## 4. Generalized Formulations of Conservation Equations

### 4.1. Exterior Calculus Formulations

We now examine generalized forms of the Reynolds transport theorem, which can be interpreted more broadly as *transformation theorems*, i.e., they provide a continuous mapping within a domain, described by the maximal integral curves of a vector or tensor field defined with respect to a parameter space. Traditionally, this is used to define a mapping between positions connected by a velocity field, parameterized by time, to give the usual Reynolds transport theorem [1]. However, as has been shown [23,24], this is not the only possible formulation.

For maximum generality we adopt an exterior calculus formulation, e.g., [15,16,17,18,69,70,71,72,73,74,75,76]. Consider an *r*-dimensional oriented compact submanifold $\Omega^{r}$ within an *n*-dimensional orientable differentiable manifold $M^{n}$, described using a patchwork of local coordinate systems. Let V be a vector or tensor field in $M^{n}$, a function of the local coordinates and parameterized by the *m*-dimensional parameter vector C, but not a function of C. The components $C_{c}$ of C are assumed orthogonal. The field trajectories (tangent bundles) of V define the continuous multivariate map (“flow”) $\phi^{C}:M^{n}\to M^{n}$ such that $V={\left(\partial \phi^{C}/\partial C\right)}^{⊤}$. This is linear and invertible, and can be used to map the entire submanifold. If $\omega^{r}$ is an *r*-form representing a conserved quantity, its integral over $\Omega^{r}$ can be proved to satisfy [23]: (38)$$\begin{array}{c} \begin{array}{cc} d\int_{\Omega (C)}\omega^{r}\; & =\left[\int_{\Omega (C)}L_{V}^{\left(C\right)}\omega^{r}\right]\cdot dC \end{array}\begin{array}{c} =\left[\int_{\Omega (C)}\;i_{V}^{\left(C\right)}\;d\omega^{r}+\oint_{\partial \Omega (C)}\;i_{V}^{\left(C\right)}\;\omega^{r}\right]\cdot dC=\left[\int_{\Omega (C)}i_{V}^{\left(C\right)}\;d\omega^{r}+d\left(i_{V}^{\left(C\right)}\;\omega^{r}\right)\right]\cdot dC \end{array} \end{array}$$
where *d* is the exterior derivative, ∂Ω is the submanifold boundary, “·” is the dot product, $L_{V}^{\left(C\right)}$ is a multivariate Lie derivative with respect to V over parameters C, and $i_{V}^{\left(C\right)}$ is a multivariate interior product with respect to V over parameters C. The multivariate operators provide vector extensions of their usual one-parameter definitions in exterior calculus [23], while the last step in Equation (Equation 38) invokes Stokes’ theorem, imposing a regularity condition on $\omega^{r}$ in the submanifold Ω. For C=t, Equation (Equation 38) reduces to the one-parameter exterior calculus formulation of the Reynolds transport theorem [18].

If the vector or tensor field V is also a function of C, the problem can be analyzed by augmenting the manifold with the parameter space, to define the flow ${\hat{\phi }}^{C}:M^{n}\times R^{m}\to M^{n}\times R^{m}$ based on the augmented (n+m)×m tensor field $V∟C={\left(\partial {\hat{\phi }}^{C}/\partial C\right)}^{⊤}$, where ∟ denotes this composition [23]. This is given by $V∟C=[\begin{array}{c} V\\ I_{m} \end{array}]$, where $I_{m}$ is the identity matrix of size *m*. Applying Equation (Equation 38) based on V∟C, this simplifies to the extended theorem [23]: (39)$$\begin{array}{c} \begin{array}{cc} \hat{d}\int_{\Omega (C)}\omega^{r}\; & =\left[\int_{\Omega (C)}L_{V∟C}^{\left(C\right)}\;\omega^{r}\right]\cdot dC=\left[\int_{\Omega (C)}\partial_{C}\omega^{r}+i_{V}^{\left(C\right)}\;d\omega^{r}+d\left(i_{V}^{\left(C\right)}\;\omega^{r}\right)\right]\cdot dC \end{array} \end{array}$$
where $\hat{d}$ is the extended exterior derivative based on the augmented coordinates, and $\partial_{C}$ is a vector partial derivative operator with respect to the components of C. For Cartesian parameters C, $\partial_{C}=\nabla_{C}$, while for X=x, C=t and V=u, Equation (Equation 39) reduces to the known exterior calculus formulation for a time-varying velocity field u(t) [15,18,22].

We can now extract the Lie differential equations applicable to each differential form in Ω, respectively, from Equations (Equation 38) and (Equation 39): (40)$$\begin{array}{c} \begin{array}{cc} L_{V}^{\left(C\right)}\omega^{r}\; & =i_{V}^{\left(C\right)}\;d\omega^{r}+d\left(i_{V}^{\left(C\right)}\;\omega^{r}\right) \end{array} \end{array}$$
(41)$$\begin{array}{c} \begin{array}{cc} L_{V∟C}^{\left(C\right)}\omega^{r}\; & =\partial_{C}\omega^{r}+L_{V}^{\left(C\right)}\omega^{r}=\partial_{C}\omega^{r}+i_{V}^{\left(C\right)}\;d\omega^{r}+d\left(i_{V}^{\left(C\right)}\;\omega^{r}\right) \end{array} \end{array}$$
The former is the multivariate extension of Cartan’s relation of exterior calculus e.g., [17,18,72], while the latter provides an extended form for a vector field V(C) that is a function of C.

### 4.2. Vector Calculus Formulations

Equations (Equation 38) and (Equation 39) provide very general equations applicable to submanifolds of any dimension in a manifold. For a system with global coordinates, these can be simplified to give a generalized parametric Reynolds transport theorem. Consider an *n*-dimensional compact domain Ω within an *n*-dimensional space M, described by the global Cartesian coordinates X. Let $V={\left(\partial X/\partial C\right)}^{⊤}={\left(\nabla_{C}X\right)}^{⊤}$ be a vector or tensor field in M, using the ∂(→)/∂(↓) vector derivative convention (where → and ↓ refer to row and column vectors, respectively), in general with V a function of C. Let $\omega^{n}$ be an *n*-dimensional volume form defined by:(42)$$\begin{array}{c} \omega =\psi \;dX_{1}\wedge \ldots \wedge dX_{n}=\psi \;{\mathrm{vol}}_{X}^{n} \end{array}$$
where ψ(X,C) is the density of a conserved quantity, ∧ is the wedge product and ${\mathrm{vol}}_{X}^{n}$ is the volume of an infinitesimal *n*-dimensional parallelopiped spanned by the cotangents to X. We assume ψ is continuous and continuously differentiable with respect to X and C throughout Ω. It can be shown that Equation (Equation 39) then reduces to [23]: (43)$$\begin{array}{c} \begin{array}{cc} d\int_{\Omega (C)}\psi \;d^{n}X\; & =\left[\nabla_{C}\int_{\Omega (C)}\psi \;d^{n}X\right]\cdot dC \end{array}=\left[\int_{\Omega (C)}\nabla_{C}\psi \;d^{n}X+\oint_{\partial \Omega (C)}\;\psi \;V\cdot d^{n-1}X\right]\cdot dC=\left[\int_{\Omega (C)}\left[\nabla_{C}\psi +\nabla_{X}\cdot \left(\psi \;V\right)\right]d^{n}x\right]\cdot dC \end{array}$$
where *d* is now the differential operator, $\nabla_{X}$ is the *n*-dimensional nabla operator with respect to X, $\nabla_{C}$ is the *m*-dimensional nabla operator with respect to C, $d^{n}X$ is a volume element in Ω, and $d^{n-1}X$ is a directed area element with outward unit normal on the boundary ∂Ω. For consistency with the vector derivative convention used, the divergence in Equation (Equation 43) is defined by $\nabla_{X}\cdot \left(\psi \;V\right)={\left[\nabla_{X}^{⊤}\left(\psi \;V\right)\right]}^{⊤}$.

We emphasize that Equation (Equation 43) applies to a compact domain Ω with smoothly varying densities, with coordinates X measured with respect to a fixed frame of reference. Further extensions can also be derived for moving and smoothly deforming frames of reference (control domains) [3,6,23], domains with jump discontinuities [7,8], irregular and fragmenting domains [9,10] and stochastic flows [21,22].

We now combine the exterior calculus (Equation (Equation 39)) and vector calculus (Equation (Equation 43)) formulations, to provide a generalized Lie differential equation applicable to each volume form in Ω. Using the notation in the second part of Equation (Equation 42), we rewrite the first part of Equation (Equation 39): (44)$$\begin{array}{c} \hat{d}\int_{\Omega (C)}\psi \;{\mathrm{vol}}_{X}^{n}=\left[\int_{\Omega (C)}L_{V∟C}^{\left(C\right)}\left(\psi \;{\mathrm{vol}}_{X}^{n}\right)\right]\cdot dC \end{array}$$
Since the exterior derivative of the integral is equivalent to its differential, and integration of ψ with respect to ${\mathrm{vol}}_{X}^{n}$ is identical to integration over $d^{n}X$, we also have:(45)$$\begin{array}{c} \hat{d}\int_{\Omega (C)}\psi \;{\mathrm{vol}}_{X}^{n}=d\int_{\Omega (C)}\psi \;d^{n}X \end{array}$$
Combining Equations (Equation 43)–(Equation 45) and equating integrands gives, for each volume form in Ω: (46)$$\begin{array}{c} L_{V∟C}^{\left(C\right)}\left(\psi \;{\mathrm{vol}}_{X}^{n}\right)=\left[\nabla_{C}\psi +\nabla_{X}\cdot \left(\psi \;V\right)\right]d^{n}X \end{array}$$
This invokes the fundamental lemma of the calculus of variations, thus imposing a regularity assumption on ψ and V within Ω. The generalized Reynolds transport theorem in Equation (Equation 43) thus yields a Lie differential equation in the form of Equation (Equation 46), defined using the augmented field V∟C. In some flow systems, this can be simplified to give a partial differential equation applicable to each local element $d^{n}X$.

## 5. Example Flow Systems

In the following sections we explore different choices of the coordinates X, parameters C, vector or tensor field V and generalized density ψ in Equations (Equation 43) and (Equation 46), based on the velocivolumetric description and the hierarchy of densities developed in Section 2 and Section 3. This yields 11 different case study flow systems, for the six integration paths labeled in red in Figure 5, with some choices examined for both time-independent and time-dependent systems. To establish a consistent nomenclature, each system is named using its X−C coordinates and its selected density. The domains Ω and D are assumed compact, and the components of X and C are assumed orthogonal. In all cases we consider V=V(C), but report only the *intrinsic* equations, with V measured with respect to a fixed frame of reference. We also consider only smoothly-varying fields within a compact and simply-connected domain.

### 5.1. Volumetric-Temporal Formulation (Density α)

We first consider the well-known volumetric-temporal formulation of the Reynolds transport theorem [1], based on the geometric space Ω with Cartesian coordinates X=x, time parameter C=t, intrinsic velocity vector field V=u(x,t):=∂x/∂t and generalized density ψ=α(x,t), defined from the fluid density ρ(x,t). This follows integration path ➀ in Figure 5. From Equation (Equation 43) and the definition of α in Equation (Equation 29), we obtain:(47)$$\begin{array}{c} \begin{array}{cc} \frac{dQ}{dt}=\frac{DQ}{Dt}\; & =\frac{d}{dt}\underset{\Omega (t)}{\;\int \int \int \;}\alpha \;dV=\underset{\Omega (t)}{\;\int \int \int \;}\frac{\partial \alpha }{\partial t}\;dV+\underset{\partial \Omega (t)}{∯}\alpha u\cdot ndA=\underset{\Omega (t)}{\;\int \int \int \;}\left[\frac{\partial \alpha }{\partial t}+\nabla_{x}\cdot \left(\alpha u\right)\right]dV \end{array} \end{array}$$
using $dV=d^{3}x$ and $ndA=dA=d^{2}x$, where n is the outward unit normal, dA is an area element and dA is a directed area element. Equation (Equation 47) is commonly written in terms of the substantial derivative $D/Dt=\partial /\partial t+u\cdot \nabla_{x}$, expressing the transport of the conserved quantity with the fluid volume.

To extract the differential equation, we require a local form of the left-hand side of Equation (Equation 47). Using the continuity equation:(48)$$\begin{array}{c} \begin{array}{cc} 0\; & =\frac{\partial \rho }{\partial t}+\nabla_{x}\cdot \left(\rho u\right) \end{array} \end{array}$$
a simple manipulation of Equation (Equation 47) using the local specific density $\underline{\alpha }$ in Equation (Equation 24) gives:(49)$$\begin{array}{c} \begin{array}{cc} \rho \frac{D\underline{\alpha }}{Dt}\; & =\frac{\partial (\rho \underline{\alpha })}{\partial t}+\nabla_{x}\cdot \left(\rho \underline{\alpha }u\right) \end{array} \end{array}$$
Further details of this derivation are given in [77] and Appendix C.

The integral and differential Equations (Equation 47) and (Equation 49) provide generalized forms of the standard conservation laws of fluid mechanics. In these equations, the left-hand terms are generally used as placeholders for any source-sink terms or driving forces for the conserved quantity to enter or leave the fluid volume or differential fluid element. Such equations for the seven common conserved quantities (fluid mass, chemical species mass, linear momentum, angular momentum, energy, charge and entropy) are listed in Table 2. All symbols used are listed in Table 1 (note that some minor overlaps of symbols could not be avoided). The equations given in Table 2 contain typical source-sink terms for the left-hand side of each equation. More comprehensive versions for different representations or coupled phenomena can also be derived, e.g., [2,3,4,6,77,78]. To enable dimensional comparisons, the SI units of each integral and differential equation are also included in Table 2.

To the seven conservation laws, we can also add an eighth, by assigning α(x,t) to its underlying pdf p(x|t) (strictly, following the probabilistic averaging method of Appendix A, this is achieved by assigning α(x+r,t)=p(x+r|t) in Equations (Equation 75) and (Equation 76) or $\underline{\alpha }\left(x+r,t\right)=1$ [kg^−1^] in Equation (Equation 77), producing a probabilistic convolution). By normalization ${\;\int \int \int \;}_{\Omega (t)}p\left(x|t\right)\;dV=1$, the left-hand term of the Reynolds transport theorem in Equation (Equation 47) then vanishes. In consequence, for a compactly supported continuous and continuously differentiable pdf p(x|t) [79] we obtain the differential Equation [23]:(50)$$\begin{array}{c} 0=\frac{\partial p(x|t)}{\partial t}+\nabla_{x}\cdot \left(p\left(x|t\right)\;u\right) \end{array}$$
This is the *Liouville equation* of fluid mechanics [27,29,30,31]. This and its integral form are included in Table 2. As will be shown, other Liouville equations based on different pdfs can be derived for other representations.

### 5.2. Velocimetric-Temporal Formulation (Density β)

Now consider a velocimetric-temporal formulation of the Reynolds transport theorem, based on the Eulerian velocity space D with Cartesian velocity coordinates X=u, time parameter C=t, local acceleration vector field $V=\dot{u}\left(u,t\right):=\partial u/\partial t$ and generalized density ψ=β(u,t), defined from the velocimetric fluid density Д(u,t). This follows integration path ➁ in Figure 5. Recall that this requires $\dot{u}$, β and Д to be defined in the velocimetric representation, for fluid elements of velocity u aggregated over all positions. From Equation (Equation 43) and the definition of β in Equation (Equation 30) [24]: (51)$$\begin{array}{c} \begin{array}{cc} \frac{dQ}{dt}\; & =\frac{d}{dt}\underset{D(t)}{\;\int \int \int \;}\beta \;dU=\underset{D(t)}{\;\int \int \int \;}\frac{\partial \beta }{\partial t}\;dU{+}_{\partial D(t)}\beta \dot{u}\cdot n_{B}\;dB=\underset{D(t)}{\;\int \int \int \;}\left[\frac{\partial \beta }{\partial t}+\nabla_{u}\cdot \left(\beta \dot{u}\right)\right]dU \end{array} \end{array}$$
using $dU=d^{3}u$ and $n_{B}\;dB=dB=d^{2}u$, where $n_{B}$ is the outward unit normal, dB is a velocimetric boundary element and dB is a directed velocimetric boundary element.

Using the previous manipulation (Appendix C), we recover a velocimetric analog of the continuity equation at each velocity:(52)$$\begin{array}{c} \begin{array}{cc} 0\; & =\frac{\partial }{\partial t}+\nabla_{u}\cdot \left(\dot{u}\right) \end{array} \end{array}$$
and in general the differential equation based on $\overset{˘}{\beta }$:(53)$$\begin{array}{c} \begin{array}{cc} \frac{Дd\overset{˘}{\beta }}{dt}\; & =\frac{\partial (\overset{˘}{\mathrm{Д\beta}})}{\partial t}+\nabla_{u}\cdot \left(\overset{˘}{\mathrm{Д\beta}}\dot{u}\right) \end{array} \end{array}$$

The conservation laws derived from Equations (Equation 51) and (Equation 53) are listed with their SI units in Table 3. Note that from Equations (Equation 47) and (Equation 51) (see also Figure 5), each integral on path ➁ is equal to the same rate of change as its volumetric counterpart on path ➀ (see Table 2). In contrast, the differential equations are localized by velocity rather than position, and so the source-sink terms are not immediately identifiable, but nonetheless can be written in the form of Equation (Equation 53), providing convenient placeholders for all source-sink terms. The corresponding integral and differential Liouville equations, obtained by assigning β(u,t)=p(u|t), are also listed in Table 3.

### 5.3. Velocivolumetric-Temporal Formulation (Density φ)

Now consider a velocivolumetric-temporal formulation of the Reynolds transport theorem, based on the Eulerian velocity-position space D×Ω with six-dimensional Cartesian coordinates $X=[\begin{array}{c} u\\ x \end{array}]$, time parameter C=t, composite vector field $V=\left[\begin{array}{c} \dot{u}\\ u \end{array}\right]\left(u,x,t\right):=\partial \left[\begin{array}{c} u\\ x \end{array}\right]/\partial t$ and generalized density ψ=φ(u,x,t), defined from the velocivolumetric density ζ(u,x,t). This follows the double integration path ➂–➀ or ➃–➁ in Figure 5. For path ➂–➀, from Equation (Equation 43) and the definition of φ in Equation (Equation 31): (54)$$\begin{array}{cc} \frac{dQ}{dt} & =\frac{d}{dt}\underset{\Omega (t)}{\;\int \int \int \;}\underset{D(x,t)}{\;\int \int \int \;}\varphi dUdV\\ \; & =\underset{\Omega (t)}{\;\int \int \int \;}\underset{D(x,t)}{\;\int \int \int \;}\frac{\partial \varphi }{\partial t}dUdV+\underset{\partial \Omega (t)}{∯}\underset{\partial D(x,t)}{∯}\varphi \left[\begin{array}{c} \dot{u}\\ u \end{array}\right]\cdot \left[\begin{array}{c} n_{B}\\ n \end{array}\right]dBdA\\ \; & =\underset{\Omega (t)}{\;\int \int \int \;}\underset{D(x,t)}{\;\int \int \int \;}\left[\frac{\partial \varphi }{\partial t}+\nabla_{u,x}\cdot \left(\varphi \left[\begin{array}{c} \dot{u}\\ u \end{array}\right]\right)\right]dUdV\\ \; & =\underset{\Omega (t)}{\;\int \int \int \;}\underset{D(x,t)}{\;\int \int \int \;}\left[\frac{\partial \varphi }{\partial t}+\nabla_{u}\cdot \left(\varphi \dot{u}\right)+\nabla_{x}\cdot \left(\varphi u\right)\right]dUdV \end{array}$$
The alternative path ➃–➁ can also be written using the second part of Equation (Equation 31). For separable integrals, further simplification is possible. From Equation (Equation 54) we can extract the continuity equation and differential equation based on $\underline{\overset{˘}{\varphi }}$ (Appendix C), respectively: (55)$$\begin{array}{c} \begin{array}{cc} 0\; & =\frac{\partial \zeta }{\partial t}+\nabla_{u}\cdot \left(\zeta \dot{u}\right)+\nabla_{x}\cdot \left(\zeta u\right) \end{array} \end{array}$$
(56)$$\begin{array}{cc} \zeta \frac{d\underline{\overset{˘}{\varphi }}}{dt}\; & =\frac{\partial (\zeta \underline{\overset{˘}{\varphi }})}{\partial t}+\nabla_{u}\cdot \left(\zeta \underline{\overset{˘}{\varphi }}\dot{u}\right)+\nabla_{x}\cdot \left(\zeta \underline{\overset{˘}{\varphi }}u\right) \end{array}$$

The conservation laws derived from Equations (Equation 54) and (Equation 56) are listed with their SI units in Table 4. Again the integrals equate to dQ/dt, hence to the same rates of change as the volumetric form (Table 2). The left-hand sides of the differential equations are written in the form of Equation (Equation 56). The corresponding Liouville equations, obtained from φ(u,x,t)=p(u,x|t), are also listed in Table 4.

### 5.4. Velocimetric-Temporal Formulation (Density φ)

Now consider a different velocimetric-temporal formulation, defined as in Section 5.2 but using the generalized density ψ=φ(u,x,t) based on ζ(u,x,t). This follows the partial integration path ➂ in Figure 5. From relation Equation (Equation 32 between φ and α:(57)$$\begin{array}{c} \begin{array}{c} d\underset{D(x,t)}{\;\int \int \int \;}\varphi \;dU=d\alpha =\nabla_{x}\alpha \cdot dx+\frac{\partial \alpha }{\partial t}dt \end{array} \end{array}$$
Examining the time derivative term using Equation (Equation 43): (58)$$\begin{array}{c} \begin{array}{cc} \frac{\partial \alpha }{\partial t}\; & =\frac{\partial }{\partial t}\underset{D(x,t)}{\;\int \int \int \;}\varphi \;dU=\underset{D(x,t)}{\;\int \int \int \;}\frac{\partial \varphi }{\partial t}\;dU{+}_{\partial D(x,t)}\varphi \dot{u}\cdot n_{B}\;dB=\underset{D(x,t)}{\;\int \int \int \;}\left[\frac{\partial \varphi }{\partial t}+\nabla_{u}\cdot \left(\varphi \dot{u}\right)\right]dU \end{array} \end{array}$$
For the conserved quantities considered, it is not straightforward to extract a differential equation from Equation (Equation 58). Instead, it is necessary to write a Lie differential Equation (Equation 46) in terms of the Lie derivative $L_{\dot{u}∟t}^{\left(t\right)}$ with respect to the augmented local acceleration field $\dot{u}∟t:=\left[\begin{array}{c} \dot{u}\\ 1 \end{array}\right]$. The left-hand term can then be treated as a placeholder for sources, sinks or drivers of the conserved quantity in the volume form ${\mathrm{vol}}_{u}^{3}$. The conservation laws derived from Equation (Equation 58) are listed in Table 5.

For this formulation, it is possible to derive two sets of temporal Liouville equations, by assigning the density φ(u,x,t), respectively, to p(u,x|t) or p(u|x,t). The former gives a Lie differential equation, while the latter reduces by normalization in Equation (Equation 8 to a partial differential equation. Both sets are listed in Table 5.

### 5.5. Volumetric-Temporal Formulation (Density φ)

We here consider a different volumetric-temporal formulation, defined as in Section 5.1 but using the generalized density ψ=φ(u,x,t) based on ζ(u,x,t). This follows the partial integration path ➃ in Figure 5. From Equation (Equation 33) between φ and β:(59)$$\begin{array}{c} \begin{array}{c} d\underset{\Omega (u,t)}{\;\int \int \int \;}\varphi \;dV=d\beta =\nabla_{u}\beta \cdot du+\frac{\partial \beta }{\partial t}dt \end{array} \end{array}$$
Examining the time derivative term using Equation (Equation 43): (60)$$\begin{array}{c} \begin{array}{cc} \frac{\partial \beta }{\partial t}\; & =\frac{\partial }{\partial t}\underset{\Omega (u,t)}{\;\int \int \int \;}\varphi \;dV=\underset{\Omega (u,t)}{\;\int \int \int \;}\frac{\partial \varphi }{\partial t}\;dV{+}_{\underset{\partial \Omega (t)}{∯}}\varphi u\cdot n\;dA=\underset{\Omega (u,t)}{\;\int \int \int \;}\left[\frac{\partial \varphi }{\partial t}+\nabla_{x}\cdot \left(\varphi u\right)\right]dV \end{array} \end{array}$$
We again extract Lie differential Equation (Equation 46), here written in terms of the Lie derivative $L_{u∟t}^{\left(t\right)}$ with respect to the augmented velocity field $u∟t:=[\begin{array}{c} u\\ 1 \end{array}]$. The conservation laws derived from Equation (Equation 60) are listed in Table 6. We also list two sets of temporal Liouville equations based on p(u,x|t) or p(x|u,t).

### 5.6. Velocimetric-Spatial (Time-Independent) Formulation (Density φ)

Now consider a time-independent velocimetric-spatial formulation of the Reynolds transport theorem, based on the velocity space D with Cartesian velocity coordinates X=u, position parameter vector C=x, velocity gradient tensor field $V={\left(G\left(u,x\right)\right)}^{⊤}:={\left(\nabla_{x}u\right)}^{⊤}$ and generalized density ψ=φ(u,x), defined from the velocivolumetric fluid density ζ(u,x). This follows integration path ➂ in Figure 5, but is independent of time, representing a stationary flow system (this formulation can also be applied to statistically stationary flow systems—for example a turbulent flow at steady state—by mapping the time dependence to a velocity dependence, thus φ(u,x,t)↦φ(u,x,t(u))). For the spatial gradient term in Equation (Equation 57), from Equation (Equation 43) and relation (Equation 32) between φ and α [23,24]:
(61)$$\begin{array}{c} \begin{array}{cc} \nabla_{x}\alpha \; & =\nabla_{x}\underset{D(x)}{\;\int \int \int \;}\varphi \;dU=\underset{D(x)}{\;\int \int \int \;}\nabla_{x}\varphi \;dU{+}_{\partial D(x)}\varphi G^{⊤}\cdot n_{B}\;dB=\underset{D(x)}{\;\int \int \int \;}\left[\nabla_{x}\varphi +\nabla_{u}\cdot \left(\varphi G^{⊤}\right)\right]dU \end{array} \end{array}$$
This gives an integral equation for the spatial gradient of the volumetric density α. We can also write a Lie differential Equation (Equation 46), expressed in terms of the multivariate Lie derivative $L_{G^{⊤}∟x}^{\left(x\right)}$ with respect to the augmented field $G∟x:=\nabla_{x}\left[u,x\right]$ over parameters x. The conservation laws derived from Equation (Equation 61) are listed in Table 7. We also list two sets of spatial Liouville equations based on p(u,x) or p(u|x) [23].

### 5.7. Volumetric-Velocital (Time-Independent) Formulation (Density φ)

Now consider a time-independent volumetric-velocital formulation of the Reynolds transport theorem, based on the geometric space Ω with Cartesian coordinates X=x, velocity parameter vector C=u, inverse velocity gradient tensor field $V={\left(\Gamma \left(u,x\right)\right)}^{⊤}:={\left(\nabla_{u}x\right)}^{⊤}$ and generalized density ψ=φ(u,x) defined from ζ(u,x). (The descriptor *velocital*, from Latin *velocitas* and -*al* for “pertaining to velocity”, follows [80].) This formulation follows integration path ➃ in Figure 5; recall that this adopts the velocimetric representation, which is integrated over the volumetric space for a distinct velocity. We again consider the time-independent case. From Equations (Equation 43), (Equation 59) and relation (Equation 33) between φ and β: (62)$$\begin{array}{c} \begin{array}{cc} \nabla_{u}\beta \; & =\nabla_{u}\underset{\Omega (u)}{\;\int \int \int \;}\varphi \;dV=\underset{\Omega (u)}{\;\int \int \int \;}\nabla_{u}\varphi \;dV{+}_{\partial \Omega (u)}\varphi \Gamma^{⊤}\cdot n\;dA=\underset{\Omega (u)}{\;\int \int \int \;}\left[\nabla_{u}\varphi +\nabla_{x}\cdot \left(\varphi \Gamma^{⊤}\right)\right]dV \end{array} \end{array}$$
This gives the gradient in velocity space of the velocimetric density β. The conservation laws derived from Equation (Equation 62) are listed in Table 8. We also list two sets of spatial Liouville equations based on p(u,x) or p(x|u).

### 5.8. Velocimetric-Spatiotemporal Formulation (Density φ)

Now consider the complete velocimetric-spatiotemporal formulation of the Reynolds transport theorem, a time-dependent extension of Section 5.6 (path ➂ in Figure 5). This examines the velocity space D with X=u, $C={\left[x,t\right]}^{⊤}$, $V={\left(\tilde{G}\left(u,x,t\right)\right)}^{⊤}:={\left({▾}_{x}u\right)}^{⊤}$, where ${▾}_{x}=\nabla_{x,t}$ is the spatiotemporal gradient, and density ψ=φ(u,x,t) defined from ζ(u,x,t). Here $\tilde{G}={\left[G,\dot{u}\right]}^{⊤}$ can be recognized as the velocity gradient tensor field G augmented with the local acceleration vector field $\dot{u}:=\partial u/\partial t$. From Equations (Equation 43 and (Equation 32 [23]: (63)$$\begin{array}{c} \begin{array}{cc} {▾}_{x}\alpha \; & ={▾}_{x}\underset{D(x,t)}{\;\int \int \int \;}\varphi \;dU=\underset{D(x,t)}{\;\int \int \int \;}{▾}_{x}\varphi \;dU{+}_{\partial D(x,t)}\varphi {\tilde{G}}^{⊤}\cdot n_{B}\;dB=\underset{D(x,t)}{\;\int \int \int \;}\left[{▾}_{x}\varphi +\nabla_{u}\cdot \left(\varphi {\tilde{G}}^{⊤}\right)\right]dU \end{array} \end{array}$$
This gives the spatiotemporal gradient ${▾}_{x}\alpha ={\left[\nabla_{x}\alpha ,\dot{\alpha }\right]}^{⊤}$ of the volumetric density α. The conservation laws derived from Equation (Equation 63) are listed in Table 9. We also present two sets of composite Liouville equations based on p(u,x|t) or p(u|x,t).

### 5.9. Volumetric-Velocitemporal Formulation (Density φ)

Now consider the complete volumetric-velocitemporal formulation of the Reynolds transport theorem, a time-dependent extension of that in Section 5.7 (path ➃ in Figure 5). This examines the geometric space Ω with X=x, $C={\left[u,t\right]}^{⊤}$, $V={\left(\tilde{\Gamma }\left(u,x,t\right)\right)}^{⊤}:={\left({▾}_{u}x\right)}^{⊤}$, where ${▾}_{u}=\nabla_{u,t}$ is a velocitemporal gradient operator, and density ψ=φ(u,x,t) defined from ζ(u,x,t). We recognize $\tilde{\Gamma }={\left[\Gamma ,u\right]}^{⊤}$ as the inverse velocity gradient tensor field Γ augmented with the local velocity vector field u:=∂x/∂t. From Equations (Equation 43 and (Equation 33):(64)$$\begin{array}{c} \begin{array}{cc} {▾}_{u}\beta \; & ={▾}_{u}\underset{\Omega (u,t)}{\;\int \int \int \;}\varphi \;dV=\underset{\Omega (u,t)}{\;\int \int \int \;}{▾}_{u}\varphi \;dV{+}_{\partial \Omega (u,t)}\varphi {\tilde{\Gamma }}^{⊤}\cdot n\;dA=\underset{\Omega (u,t)}{\;\int \int \int \;}\left[{▾}_{u}\varphi +\nabla_{x}\cdot \left(\varphi {\tilde{\Gamma }}^{⊤}\right)\right]dV \end{array} \end{array}$$
This provides the velocitemporal gradient ${▾}_{u}\beta ={\left[\nabla_{u}\beta ,\dot{\beta }\right]}^{⊤}$ of the velocimetric density β. The conservation laws derived from Equation (Equation 64) are listed in Table 10. We also list two sets of composite Liouville equations, here based on p(u,x|t) or p(x|u,t).

### 5.10. Velocimetric-Temporal Formulation (Density θ)

Now consider the alternative velocimetric-temporal formulation on path ➄ in Figure 5, based on the generalized density ψ=θ(u,x,t) defined from the fluid density η(u,x,t). From the definition of θ in Equations (Equation 34) and (Equation 36):(65)$$\begin{array}{c} \begin{array}{c} d\underset{D(x,t)}{\;\int \int \int \;}\theta \;dU=dQ\left(t\right)=\frac{dQ}{dt}dt \end{array} \end{array}$$
Note the peculiar property that integrating θ(u,x,t) with respect to u also eliminates x, giving a velocimetric-temporal formulation with x=u, C=t and $V=\dot{u}$. From Equation (Equation 43):(66)$$\begin{array}{c} \begin{array}{cc} \frac{dQ}{dt}=\frac{d}{dt}\underset{D(x,t)}{\;\int \int \int \;}\theta \;dU\; & =\underset{D(x,t)}{\;\int \int \int \;}\left[\frac{\partial \theta }{\partial t}+\nabla_{u}\cdot \left(\theta \dot{u}\right)\right]dU \end{array} \end{array}$$
The left-hand terms are again equivalent to those in Table 2, Table 3 and Table 4. The extracted continuity equation and differential equation based on $\overset{˘}{\underline{\theta }}$ (Appendix C) are: (67)$$\begin{array}{c} \begin{array}{cc} 0\; & =\frac{\partial \eta }{\partial t}+\nabla_{u}\cdot \left(\eta \dot{u}\right) \end{array} \end{array}$$
(68)$$\begin{array}{c} \begin{array}{cc} \eta \frac{d\overset{˘}{\underline{\theta }}}{dt}\; & =\frac{\partial (\eta \overset{˘}{\underline{\theta }})}{\partial t}+\nabla_{u}\cdot \left(\eta \overset{˘}{\underline{\theta }}\dot{u}\right) \end{array} \end{array}$$
The conservation laws derived from Equations (Equation 66) and (Equation 68) are listed in Table 11. Apart from the different fluid density η and specific density labels, these are mathematically identical to the conservation laws based on the velocimetric density Д in Table 3. The Liouville equations based on equating θ to its underlying pdf p(u|x,t) are also listed, giving an alternative route to these equations to that given in Table 5.

### 5.11. Volumetric-Temporal Formulation (Density ϵ)

Now consider the alternative volumetric-temporal formulation on path ⑥ in Figure 5, based on the generalized density ψ=ϵ(u,x,t) defined from the fluid density ξ(u,x,t). From the definition of ϵ in Equations (Equation 35) and (Equation 37):(69)$$\begin{array}{c} \begin{array}{c} d\underset{\Omega (u,t)}{\;\int \int \int \;}\epsilon \;dV=dQ\left(t\right)=\frac{dQ}{dt}dt \end{array} \end{array}$$
This also has the peculiar property that integration with respect to x eliminates u, reducing this to a volumetric-temporal formulation with X=x, C=t and V=u. From Equation (Equation 43):(70)$$\begin{array}{c} \begin{array}{cc} \frac{dQ}{dt}=\frac{d}{dt}\underset{\Omega (u,t)}{\;\int \int \int \;}\epsilon \;dV\; & =\underset{\Omega (u,t)}{\;\int \int \int \;}\left[\frac{\partial \epsilon }{\partial t}+\nabla_{x}\cdot \left(\epsilon u\right)\right]dV \end{array} \end{array}$$
The left-hand terms are again equivalent to those in Table 2, Table 3 and Table 4. The extracted continuity equation and differential equation based on $\overset{˘}{\underline{\epsilon }}$ (Appendix C) are: (71)$$\begin{array}{c} \begin{array}{cc} 0\; & =\frac{\partial \xi }{\partial t}+\nabla_{x}\cdot \left(\xi u\right) \end{array} \end{array}$$
(72)$$\begin{array}{c} \begin{array}{cc} \xi \frac{d\overset{˘}{\underline{\epsilon }}}{dt}\; & =\frac{\partial (\xi \overset{˘}{\underline{\epsilon }})}{\partial t}+\nabla_{x}\cdot \left(\xi \overset{˘}{\underline{\epsilon }}u\right) \end{array} \end{array}$$
The conservation laws derived from Equations (Equation 70) and (Equation 72) are listed in Table 12. Apart from the fluid density ξ and specific density labels, these are mathematically identical to the volumetric conservation laws based on ρ in Table 2. The Liouville equations based on the underlying pdf p(x|u,t) are also listed, providing an alternative route to that in Table 6.

### 5.12. Discussion

Comparing the above analyses, we see that the first three and last two formulations in Section 5.1, Section 5.2 and Section 5.3 and Section 5.10 and Section 5.11 are connected by their equivalence to the rate of change of the conserved quantity dQ/dt, and thence to the source-sink terms of the standard (volumetric) integral equations of fluid mechanics. These are based on complete integrations, respectively, of the densities α, β, φ, θ and ϵ.

The remaining examples involve partial integrations of the velocivolumetric density φ. Of these, the velocimetric-spatiotemporal formulation in Section 5.8 combines the velocimetric-temporal and velocimetric-spatial formulations in Section 5.4 and Section 5.6, connected by Equation (Equation 57), based on α and φ. Similarly, the volumetric-velocitemporal formulation in Section 5.9 combines the volumetric-temporal and volumetric-velocital formulations in Section 5.5 and Section 5.7, connected by Equation (Equation 59), based on β and φ. For these formulations, it is possible to extract a Lie differential equation based on a Lie operator specific to that formulation.

All formulations can be used to derive a number of Liouville equations based on the underlying pdfs defined in Section 3.1 and Figure 2. In general, these take the form of Lie differential equations, but for cases in which the pdf is normalized by the integration, the analysis yields a homogeneous integral equation and associated partial differential equation.

## 6. Conclusions

This study examines a generalized framework for the Reynolds transport theorem [23,24], which enables continuous multiparametric mappings of a differential form on a manifold—or of a density within a generalized coordinate space—connected by the maximal integral curves or surfaces described by a vector or tensor field. These extend the formulation of [1] to encompass new *transformation theorems*, which exploit previously unreported multiparametric continuous (Lie) symmetries of a vector or tensor field associated with a conserved quantity. In this study, we explore the implications of the generalized framework for fluid flow systems, using an extended Eulerian velocivolumetric description of fluid flow in place of the standard Eulerian description.

The analysis commences in Section 2 with a detailed discussion of the extended velocity-position description, and of the form and connection between the geometric and velocity domains for different classes of fluid flow systems. In Section 3.1, we then define a hierarchy of five pdfs {p(x|t),p(u|t),p(u,x|t),p(u|x,t),p(x|u,t)} within this description, the properties of which are then explored. In Section 3.2 and Section 3.3, these are used by convolution to define analogous hierarchies of fluid densities {ρ,Д,ζ,η,ξ} and generalized densities {α,β,φ,θ,ϵ}, of which only the first (ρ and α) are commonly known. The generalized framework for the Reynolds transport theorem is presented in Section 4, in both exterior calculus and vector calculus formulations. Its connection to underlying partial differential equations and Lie differential equations—the latter containing the Lie derivative of a volume form in the domain—is also examined. In Section 5, the densities and theorems are used to obtain 11 formulations of the Reynolds transport theorem arising from the veloci-volumetric description, for different choices of the coordinate space, parameter space and density. These are reported in the form of 11 tables of integral and differential conservation laws applicable to these systems, for the eight common conserved quantities of interest in fluid mechanics (fluid mass, species mass, linear momentum, angular momentum, energy, charge, entropy and probability). The equations for conservation of probability can be interpreted as analogs of the Liouville equation, applicable to different spaces. The analyses provide a considerable assortment of new conservation laws for the analysis of fluid flow systems.

While every effort has been made to provide a comprehensive treatment, this study exclusively considers compact domains with Cartesian position and velocity coordinates, and no attempt is made to examine orthogonal or non-orthogonal curvilinear coordinate systems or parameter spaces (however, care is taken to distinguish the vector of partial derivatives $\partial_{x}$ from the gradient $\nabla_{x}$). Further work is required to extend these analyses to more general domains and coordinate systems. Further detailed study is also required of the partial and Lie differential equations reported in Table 3, Table 4, Table 5, Table 6, Table 7, Table 8, Table 9, Table 10, Table 11 and Table 12, to identify the appropriate source-sink terms for each expression. The analyses are also restricted to fixed frames of reference (control volumes or domains), but can readily be extended to moving and deforming frames of reference using relative vector or tensor fields $V_{rel}$, e.g., [3,6,23]. The analyses could also be extended to consider domains with jump discontinuities [7,8], irregular and fragmenting domains [9,10], or special or general relativity [81]. The generalized Reynolds theorem framework can also be used to generate conservation laws for other dynamical systems containing conserved quantities [23].

## Figures and Tables

**Figure 1 entropy-24-01493-f001:**
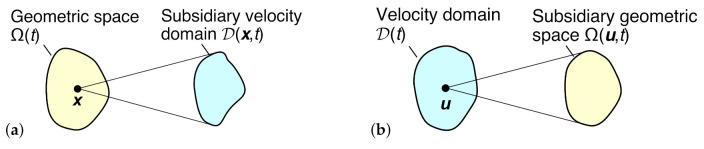
Schematic diagrams of the mapping between domains Ω and D in (**a**) the geometric representation, and (**b**) the velocimetric representation.

**Figure 2 entropy-24-01493-f002:**
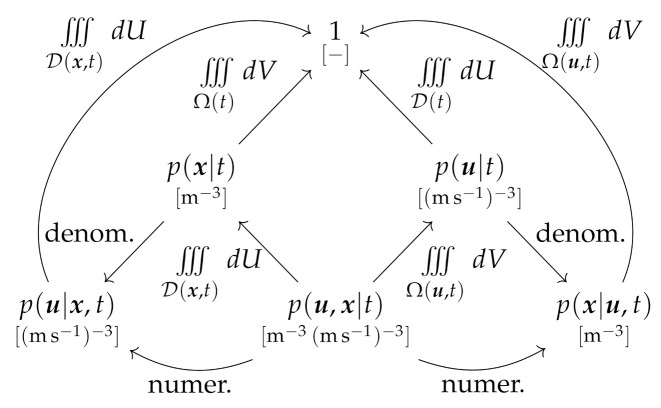
Relational diagram between the pdfs defined in this study.

**Figure 3 entropy-24-01493-f003:**
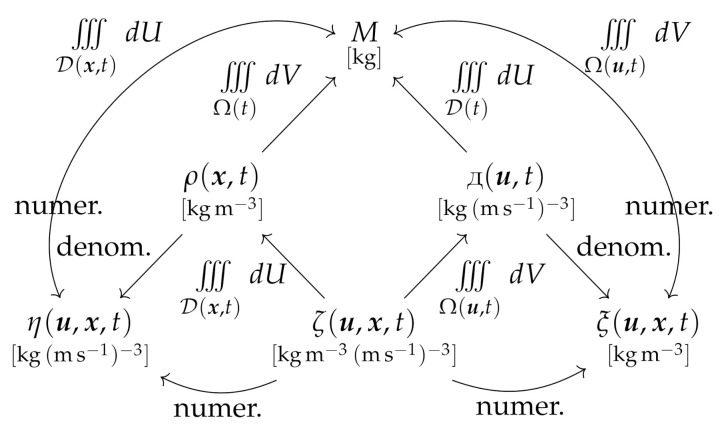
Relational diagram between the fluid or material densities defined in this study.

**Figure 4 entropy-24-01493-f004:**
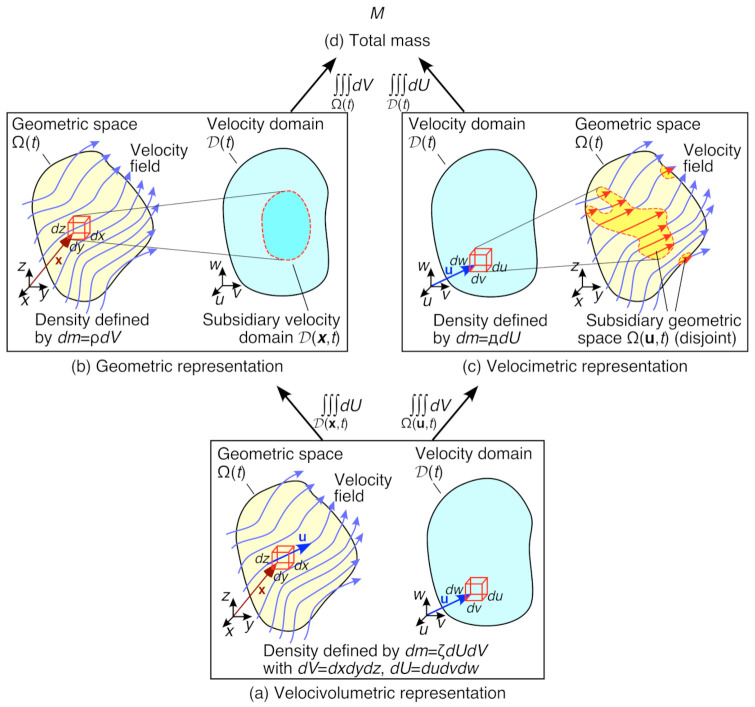
Schematic diagrams for the three major fluid or material densities of this study: (**a**) the velocivolumetric density ζ in the velocivolumetric, (**b**) the volumetric density ρ in the geometric representation, (**c**) the velocimetric density Д in the velocimetric representation, and (**d**) the total mass *M*. These are drawn using the commutative diagram format given in Figure 3.

**Figure 5 entropy-24-01493-f005:**
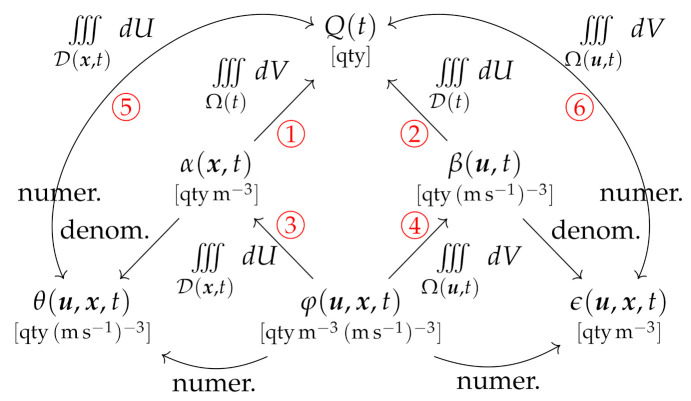
Relational diagram between the generalized densities defined in this study (the integration paths are numbered in red).

**Table 1 entropy-24-01493-t001:** Nomenclature used in this study.

Symbol	Description	SI Units
**Mathematical Operators**		
${}^{⊤}$	transpose	
·	vector scalar product	
×	cross product; multiplication symbol (only if there is a line break in the equation)	
*∂*	partial derivative operator; boundary of domain	
$\partial_{C}$	vector partial derivative operator with respect to C	
$\nabla_{C}$	gradient operator with respect to C	
$\nabla_{u}$	velocital gradient operator with respect to u	(m s^−1^)^−1^
$\nabla_{x}$	spatial gradient operator with respect to x	m^−1^
$\nabla_{X}$	gradient operator with respect to X	
${▾}_{u}=\nabla_{u,t}$	velocitotemporal gradient operator with respect to [u,t]	[(m s^−1^)^−1^, (m s^−1^)^−1^, (m s^−1^)^−1^, s^−1^]
${▾}_{x}=\nabla_{x,t}$	spatiotemporal gradient operator with respect to [x,t]	[m^−1^, m^−1^, m^−1^, s^−1^]
[·]	expectation over small volume U	
〈·〉	expectation over small velocity domain V	
[[[·]]]	integral over volume Ω	
〈〈〈·〉〉〉	integral over velocity domain D	
∧	wedge product	
∟	augmentation operator, such that V∟C is the tensor V based on coordinates X augmented by parameter C	
➀	integral path as labeled in Figure 5	
**Conventions**		
Vector derivatives are defined by the ∂(→)/∂(↓) convention		
The product of two vectors implies a tensor, e.g., $uu:=uu^{⊤}$		
The divergence of a tensor is rotated, e.g., $\nabla_{x}\cdot G:={\left(\nabla_{x}^{⊤}G\right)}^{⊤}$		
**Roman symbols**		
*c*	index of chemical species; index of components of C	
$C_{c}$	*c*th component of C	
C	generalized *m*-dimensional parameter vector	
CV	control volume = reference frame for fluid motion	
*d*	differential of a function; exterior derivative of a differential form	
$\hat{d}$	extended exterior derivative based on augmented coordinates	
d/dt	total derivative in time	s^−1^
dA	infinitesimal area element in volumetric space	m^2^
dA	directed infinitesimal area element in volumetric space	m^2^
dB	infinitesimal surface element in velocimetric space	(m s^−1^)^2^
dB	directed infinitesimal surface element in velocimetric space	(m s^−1^)^2^
dm	infinitesimal element of fluid mass	kg
dU=dudvdw	infinitesimal element of velocimetric space	(m s^−1^)^3^
dV=dxdydz	infinitesimal element of volumetric space	m^3^
$dX_{j}$	cotangent to *j*th component of generalized vector X	
$d^{n-1}X$	generalized directed area element on ∂Ω	
$d^{n}X$	generalized volume element in Ω	
$d_{x,t}$	vector of spatiotemporal SI units	[m, m, m, s]
$d_{u,t}$	vector of velocitotemporal SI units	[m s^−1^, m s^−1^, m s^−1^, s]
D/Dt	substantial or material derivative in time	s^−1^
D	velocimetric domain	
$\underline{e}$	local specific total energy	J kg^−1^
$\overset{˘}{e}$	velocity-distinct specific energy	J kg^−1^
$\overset{˘}{\underline{e}}$	velocity-distinct local specific energy	J kg^−1^
*E*	total energy	J
∑F	sum of forces	N
g	acceleration due to gravity	m s^−2^
$G:=\nabla_{x}u$	velocity gradient tensor field	(m s^−1^) m^−1^ = s^−1^
$\tilde{G}:={▾}_{x}u$	augmented velocity gradient tensor field	[s^−1^, s^−1^, s^−1^, m s^−2^]
i	electrical flux	C m^−2^ s^−1^ = A m^−2^
$i_{V}^{\left(C\right)}$	multivariate interior product with respect to V over parameters C	
*I*	net inward electrical current (passive sign convention)	C s^−1^ = A
$I_{m}$	identity matrix of size *m*	
*j*	index of components of generalized coordinates X	
$j_{c}$	molar flux of species *c*	mol m^−2^ s^−1^
$j_{Q}$	heat flux	J m^−2^ s^−1^
$j_{S}$	entropy flux	J K^−1^ m^−2^ s^−1^
$L_{V}^{\left(C\right)}$	multivariate Lie derivative with respect to V over parameters C	
$L_{V∟C}^{\left(C\right)}$	multivariate Lie derivative with respect to V∟C over parameters C	
LHS	left-hand side	
m	small fluid mass domain	
*m*	dimension of vector parameter C	
M	orientable differentiable manifold; generalized space	
*M*	total fluid mass	kg
$\dot{M}$	rate of change of total fluid mass	kg s^−1^
${\dot{M}}_{c}$	rate of change of mass of species *c*	kg_c_ s^−1^
$M_{c}$	molar mass of species *c*	kg_c_ mol^−1^	
*n*	dimension of manifold *M*, dimension of coordinates X	
n	outward unit normal to Ω in volumetric space	
$n_{B}$	outward unit normal to D in velocimetric space	
$p\left(a,b|c\right)=p_{a,b|c}$	conditional pdf of *a* and *b* subject to *c*	units of ${\left(ab\right)}^{-1}$
*P*	pressure	Pa = J m^−3^
*Q*	total conserved quantity (of any type)	qty
${\dot{Q}}_{in}$	net inward heat flow rate	J s^−1^
*r*	dimension of submanifold Ω, dimension of differential form $\omega^{r}$	
$r={\left[r_{x},r_{y},r_{z}\right]}^{⊤}$	local Cartesian position coordinates	m
$\underline{r}$	local radius of a lever arm	m
$\overset{˘}{r}$	velocity-distinct radius of a lever arm	m
$\underline{\overset{˘}{r}}$	velocity-distinct local radius of a lever arm	m
RHS	right-hand side	
$\underline{s}$	local specific entropy	J K^−1^ kg^−1^
$\overset{˘}{s}$	velocity-distinct specific entropy	J K^−1^ kg^−1^
$\overset{˘}{\underline{s}}$	velocity-distinct local specific entropy	J K^−1^ kg^−1^
$s={\left[s_{u},s_{v},s_{w}\right]}^{⊤}$	local Cartesian velocity coordinates	m s^−1^
*S*	total entropy	J K^−1^
${\dot{S}}^{nf}$	total net inward non-fluid entropy flow rate	J K^−1^ s^−1^
*t*	time	s
∑T	sum of torques	N m
$u={\left[u,v,w\right]}^{⊤}$	Cartesian velocity field $:=\frac{\partial x}{\partial t}$	m s^−1^
$\dot{u}={\left[\dot{u},\dot{v},\dot{w}\right]}^{⊤}$	Cartesian local acceleration field $:=\frac{\partial u}{\partial t}$	m s^−2^
U	small velocity domain	
${\mathrm{vol}}_{X}^{n}$	volume of an infinitesimal *n*-dimensional parallelopiped spanned by the cotangents to X	
$V_{jc}$	(jc)th component of generalized vector or tensor field V	
V	generalized vector or tensor field	
V	small fluid volume	
${\dot{W}}_{in}$	net inward work flow rate	J s^−1^
$x={\left[x,y,z\right]}^{⊤}$	Cartesian position coordinates	m
$x_{0}={\left[x_{0},y_{0},z_{0}\right]}^{⊤}$	Cartesian Lagrangian position coordinates	m
$X_{j}$	*j*th component of vector X	
X	generalized *n*-dimensional local or global Cartesian coordinates	
$z_{c}$	charge per mass of species *c*	C kg_c_^−1^
$\underline{z}$	local specific charge	C kg^−1^
$\overset{˘}{z}$	velocity-distinct specific charge	C kg^−1^
$\overset{˘}{\underline{z}}$	velocity-distinct local specific charge	C kg^−1^
*Z*	total charge	C
**Greek symbols**		
$\alpha =\rho \;\underline{\alpha }$	generalized volumetric density	qty m^−3^
$\underline{\alpha }$	local generalized specific density	qty kg^−1^
$\beta =Д\;\overset{˘}{\beta }$	generalized velocimetric density	qty (m s^−1^)^−3^
$\overset{˘}{\beta }$	velocity-distinct generalized specific density	qty kg^−1^
$\Gamma :=\nabla_{u}x$	inverse velocity gradient tensor field	m (m s^−1^)^−1^ = s
$\tilde{\Gamma }:={▾}_{u}x$	augmented inverse velocity gradient tensor field,	[s, s, s, m s^−1^]
δ	Kronecker delta tensor	
$\epsilon =\xi \;\underline{\overset{˘}{\epsilon }}$	generalized conditional volumetric density	qty m^−3^
$\underline{\overset{˘}{\epsilon }}$	velocity-distinct local generalized specific density	qty kg^−1^
ζ	velocivolumetric fluid mass density	kg m^−3^ (m s^−1^)^−3^
$\zeta_{c}$	velocivolumetric mass density of species *c*	kg_c_ m^−3^ (m s^−1^)^−3^
η	conditional velocimetric fluid mass density	kg (m s^−1^)^−3^
$\eta_{c}$	conditional velocimetric mass density of species *c*	kg_c_ (m s^−1^)^−3^
$\theta =\eta \;\underline{\overset{˘}{\theta }}$	generalized conditional velocimetric density	qty (m s^−1^)^−3^
$\underline{\overset{˘}{\theta }}$	velocity-distinct local generalized specific density	qty kg^−1^
ξ	conditional volumetric fluid mass density	kg m^−3^
$\xi_{c}$	conditional volumetric mass density of species *c*	kg_c_ m^−3^
${\hat{\dot{\xi }}}_{c}$	molar rate of production of species *c*	mol m^−3^ s^−1^
ρ	volumetric fluid mass density	kg m^−3^
$\rho_{c}$	volumetric mass density of species *c*	kg_c_ m^−3^
$\dot{\sigma }$	total entropy production	J K^−1^ s^−1^
$\hat{\dot{\sigma }}$	local entropy production	J K^−1^ m^−3^ s^−1^
τ	stress tensor (positive in compression)	Pa = J m^−3^
$\phi^{C}$	multivariate flow generated by V	
${\hat{\phi }}^{C}$	augmented multivariate flow generated by V∟C	
$\varphi =\zeta \;\underline{\overset{˘}{\varphi }}$	generalized velocivolumetric density	qty m^−3^ (m s^−1^)^−3^
$\underline{\overset{˘}{\varphi }}$	velocity-distinct local generalized specific density	qty kg^−1^
${\underline{\chi }}_{c}$	local specific mass density of species *c*	kg_c_ kg^−1^
${\overset{˘}{\chi }}_{c}$	velocity-distinct specific mass density of species *c*	kg_c_ kg^−1^
${\overset{˘}{\underline{\chi }}}_{c}$	velocity-distinct local specific mass density of species *c*	kg_c_ kg^−1^
ψ	generalized density of conserved quantity in generalized space	
$\omega^{r},\omega^{n}$	*r*-form, *n*-form (respectively) in submanifold Ω	
Ω	general submanifold or domain; volumetric domain (fluid volume or material volume)	
**Cyrillic symbols**		
Д	velocimetric fluid mass density	kg (m s^−1^)^−3^
${Д}_{c}$	velocimetric mass density of species *c*	kg_c_ (m s^−1^)^−3^

**Table 2 entropy-24-01493-t002:** Conservation Laws for the (Well-Known) Volumetric-Temporal Formulation (based on the Volumetric Fluid Density ρ(x,t)) after [2,3,4,6,77,78].

Conserved Quantity	Density	Integral Equation			Differential Equation		
	α(x,t) $=\rho \underline{\alpha }$	LHS = $\frac{dQ}{dt}$ = $\frac{DQ}{Dt}$	= RHS	SI Units	LHS	= RHS	SI Units
Fluid mass	ρ	$0=\dot{M}$ $=\frac{d[\;[\;[\rho ]\;]\;]}{dt}$	$=\underset{\Omega (t)}{\;\int \int \int \;}\left[\frac{\partial \rho }{\partial t}+\nabla_{x}\cdot \left(\rho u\right)\right]dV$	[kg s^−1^]	0	$=\frac{\partial \rho }{\partial t}+\nabla_{x}\cdot \left(\rho u\right)$	[kg s^−1^ m^−3^]
Species mass	$\rho {\underline{\chi }}_{c}$ $=\rho_{c}$	$\left({\dot{M}}_{c}\right)=\frac{d[\;\left[\;\left[\rho {\underline{\chi }}_{c}\right]\;\right]\;]}{dt}$	$=\underset{\Omega (t)}{\;\int \int \int \;}\left[\frac{\partial \rho {\underline{\chi }}_{c}}{\partial t}+\nabla_{x}\cdot \left(\rho {\underline{\chi }}_{c}u\right)\right]dV$	[kg_c_ s^−1^]	$\rho \frac{D{\underline{\chi }}_{c}}{Dt}=M_{c}\left({\hat{\dot{\xi }}}_{c}-\nabla_{x}\cdot j_{c}\right)$	$=\frac{\partial \rho {\underline{\chi }}_{c}}{\partial t}$ $+\nabla_{x}\cdot \left(\rho {\underline{\chi }}_{c}u\right)$	[kg_c_ s^−1^ m^−3^]
Linear momentum	ρu	$\sum F=\frac{d[\;[\;[\rho u]\;]\;]}{dt}$	$=\underset{\Omega (t)}{\;\int \int \int \;}\left[\frac{\partial \rho u}{\partial t}+\nabla_{x}\cdot \left(\rho uu\right)\right]dV$	[(kg m s^−1^) s^−1^ = N]	$\rho \frac{Du}{Dt}=-\nabla_{x}P-\nabla_{x}\cdot \tau +\rho g$	$=\frac{\partial \rho u}{\partial t}+\nabla_{x}\cdot \left(\rho uu\right)$	[N m^−3^]
Angular momentum ${}^{a}$	$\rho (\underline{r}\times u)$	$\sum T=\frac{d[\;\left[\;\left[\rho \left(\underline{r}\times u\right)\right]\;\right]\;]}{dt}$	$=\left\{\begin{array}{c} \underset{\Omega (t)}{\;\int \int \int \;}[\frac{\partial \rho (\underline{r}\times u)}{\partial t}\\ +\nabla_{x}\cdot \left(\rho \left(\underline{r}\times u\right)u\right)]dV \end{array}\right.$	[(kg m${}^{2}$ s^−1^) s^−1^ = N m]	$\left.\begin{array}{c} \rho \frac{D(\underline{r}\times u)}{Dt}=-\nabla_{x}\cdot \left(\underline{r}\times P\delta \right)\\ -\nabla_{x}\cdot \left(\underline{r}\times \tau \right)+\left(\underline{r}\times \rho g\right) \end{array}\right\}$	$=\left\{\begin{array}{c} \frac{\partial \rho (\underline{r}\times u)}{\partial t}\\ +\nabla_{x}\cdot \left(\rho \left(\underline{r}\times u\right)u\right) \end{array}\right.$	[N m^−2^ = (N m) m^−3^]
Energy	$\rho \underline{e}$	$\left.\begin{array}{c} \frac{DE}{Dt}=\frac{d[\;\left[\;\left[\rho \underline{e}\right]\;\right]\;]}{dt}\\ =({\dot{Q}}_{in}+{\dot{W}}_{in}) \end{array}\right\}$	$=\underset{\Omega (t)}{\;\int \int \int \;}\left[\frac{\partial \rho \underline{e}}{\partial t}+\nabla_{x}\cdot \left(\rho \underline{e}u\right)\right]dV$	[J s^−1^ = W]	$\left.\begin{array}{c} \rho \frac{D\underline{e}}{Dt}=-\nabla_{x}\cdot j_{Q}-\nabla_{x}\cdot \left(Pu\right)\\ -\nabla_{x}\cdot \left(\tau \cdot u\right)+\rho g\cdot u \end{array}\right\}$	$=\frac{\partial \rho \underline{e}}{\partial t}+\nabla_{x}\cdot \left(\rho \underline{e}u\right)$	[J s^−1^ m^−3^ = W m^−3^]
Charge (in solution)	$\rho \underline{z}$	$\left.\begin{array}{c} \frac{DZ}{Dt}=\frac{d[\;\left[\;\left[\rho \underline{z}\right]\;\right]\;]}{dt}\\ =I+\left(\sum_{c}z_{c}{\dot{M}}_{c}\right) \end{array}\right\}$	$=\underset{\Omega (t)}{\;\int \int \int \;}\left[\frac{\partial \rho \underline{z}}{\partial t}+\nabla_{x}\cdot \left(\rho \underline{z}u\right)\right]dV$	[C s^−1^ = A]	$\left.\begin{array}{c} \rho \frac{D\underline{z}}{Dt}=-\nabla_{x}\cdot i\\ +\sum_{c}M_{c}z_{c}\left({\hat{\dot{\xi }}}_{c}-\nabla_{x}\cdot j_{c}\right) \end{array}\right\}$	$=\frac{\partial \rho \underline{z}}{\partial t}+\nabla_{x}\cdot \left(\rho \underline{z}u\right)$	[C s^−1^ m^−3^ = A m^−3^]
Entropy	$\rho \underline{s}$	$\left.\begin{array}{c} \frac{DS}{Dt}=\frac{d[\;\left[\;\left[\rho \underline{s}\right]\;\right]\;]}{dt}\\ =\dot{\sigma }+{\dot{S}}^{nf} \end{array}\right\}$	$=\underset{\Omega (t)}{\;\int \int \int \;}\left[\frac{\partial \rho \underline{s}}{\partial t}+\nabla_{x}\cdot \left(\rho \underline{s}u\right)\right]dV$	[J K^−1^ s^−1^]	$\rho \frac{D\underline{s}}{Dt}=$ $\hat{\dot{\sigma }}-\nabla_{x}\cdot j_{S}$	$=\frac{\partial \rho \underline{s}}{\partial t}+\nabla_{x}\cdot \left(\rho \underline{s}u\right)$	[J K^−1^ s^−1^ m^−3^]
Probability	p(x|t) $=p_{x|t}$	$0=\frac{d[\;\left[\;\left[p_{x|t}\right]\;\right]\;]}{dt}$	$=\underset{\Omega (t)}{\;\int \int \int \;}\left[\frac{\partial p_{x|t}}{\partial t}+\nabla_{x}\cdot \left(p_{x|t}u\right)\right]dV$	[s^−1^]	0	$=\frac{\partial p_{x|t}}{\partial t}+\nabla_{x}\cdot \left(p_{x|t}u\right)$	[s^−1^ m^−3^]

^a^ It can be shown that the differential equation for angular momentum reduces to ***τ*** = ***τ***^⊤^ [3,77].

**Table 3 entropy-24-01493-t003:** Conservation Laws for the Velocimetric-Temporal Formulation (based on the Velocimetric Fluid Density Д(u,t)).

Conserved Quantity	Density	Integral Equation			Differential Equation		
	$\beta (u,t)=Д\overset{˘}{\beta }$	LHS = $\frac{dQ}{dt}$ = $\frac{DQ}{Dt}$	= RHS	SI Units	LHS	= RHS	SI Units
Fluid mass	Д	$0=\dot{M}$ $=\frac{d\langle \;\langle \;\langle Д\rangle \;\rangle \;\rangle }{dt}$	$=\underset{D(t)}{\;\int \int \int \;}\left[\frac{\partial }{\partial t}+\nabla_{u}\cdot \left(Д\dot{u}\right)\right]dU$	[kg s^−1^]	0	$=\frac{\partial Д}{\partial t}+\nabla_{u}\cdot \left(Д\dot{u}\right)$	[kg s^−1^ (m s^−1^)^−3^]
Species mass	$Д{\overset{˘}{\chi }}_{c}{Д}_{c}$	$\left({\dot{M}}_{c}\right)=\frac{d\langle \;\langle \;\langle Д{\overset{˘}{\chi }}_{c}\rangle \;\rangle \;\rangle }{dt}$	$=\underset{D(t)}{\;\int \int \int \;}\left[\frac{\partial Д{\overset{˘}{\chi }}_{c}}{\partial t}+\nabla_{u}\cdot \left(Д{\overset{˘}{\chi }}_{c}\dot{u}\right)\right]dU$	[kg_c_ s^−1^]	$Д\frac{d{\overset{˘}{\chi }}_{c}}{dt}$	$=\frac{\partial Д{\overset{˘}{\chi }}_{c}}{\partial t}+\nabla_{u}\cdot \left(Д{\overset{˘}{\chi }}_{c}\dot{u}\right)$	[kg_c_ s^−1^ (m s^−1^)^−3^]
Linear momentum	Дu	$\sum F=\frac{d\langle \;\langle \;\langle Дu\rangle \;\rangle \;\rangle }{dt}$	$=\underset{D(t)}{\;\int \int \int \;}\left[\frac{\partial Дu}{\partial t}+\nabla_{u}\cdot \left(Дu\dot{u}\right)\right]dU$	[(kg m s^−1^) s^−1^ = N]	$Д\frac{du}{dt}$	$=\frac{\partial Дu}{\partial t}+\nabla_{u}\cdot \left(Дu\dot{u}\right)$	[N (m s^−1^)^−3^]
Angular momentum	$Д(\overset{˘}{r}\times u)$	$\sum T=\frac{d\langle \;\langle \;\langle Д(\overset{˘}{r}\times u)\rangle \;\rangle \;\rangle }{dt}$	$=\underset{D(t)}{\;\int \int \int \;}\left[\frac{\partial Д(\overset{˘}{r}\times u)}{\partial t}+\nabla_{u}\cdot \left(\left(Д\overset{˘}{r}\times u\right)\dot{u}\right)\right]dU$	[(kg m${}^{2}$ s^−1^) s^−1^ = N m]	$Д\frac{d(\overset{˘}{r}\times u)}{dt}$	$=\left\{\begin{array}{c} \frac{\partial Д(\overset{˘}{r}\times u)}{\partial t}\\ +\nabla_{u}\cdot \left(Д(\overset{˘}{r}\times u)\dot{u}\right) \end{array}\right.$	[(N m) (m s^−1^)^−3^]
Energy	$Д\overset{˘}{e}$	$\left.\begin{array}{c} \frac{DE}{Dt}=\frac{d\langle \;\langle \;\langle Д\overset{˘}{e}\rangle \;\rangle \;\rangle }{dt}\\ =({\dot{Q}}_{in}+{\dot{W}}_{in}) \end{array}\right\}$	$=\underset{D(t)}{\;\int \int \int \;}\left[\frac{\partial Д\overset{˘}{e}}{\partial t}+\nabla_{u}\cdot \left(Д\overset{˘}{e}\dot{u}\right)\right]dU$	[J s^−1^ = W]	$Д\frac{d\overset{˘}{e}}{dt}$	$=\frac{\partial Д\overset{˘}{e}}{\partial t}+\nabla_{u}\cdot \left(Д\overset{˘}{e}\dot{u}\right)$	[J s^−1^ (m s^−1^)^−3^ = W (m s^−1^)^−3^]
Charge (in solution)	$Д\overset{˘}{z}$	$\left.\begin{array}{c} \frac{DZ}{Dt}=\frac{d\langle \;\langle \;\langle Д\overset{˘}{z}\rangle \;\rangle \;\rangle }{dt}\\ =I+\left(\sum_{c}z_{c}{\dot{M}}_{c}\right) \end{array}\right\}$	$=\underset{D(t)}{\;\int \int \int \;}\left[\frac{\partial Д\overset{˘}{z}}{\partial t}+\nabla_{u}\cdot \left(Д\overset{˘}{z}\dot{u}\right)\right]dU$	[C s^−1^ = A]	$Д\frac{d\overset{˘}{z}}{dt}$	$=\frac{\partial Д\overset{˘}{z}}{\partial t}+\nabla_{u}\cdot \left(Д\overset{˘}{z}\dot{u}\right)$	[C s^−1^ (m s^−1^)^−3^ = A (m s^−1^)^−3^]
Entropy	$Д\overset{˘}{s}$	$\frac{DS}{Dt}=\frac{d\langle \;\langle \;\langle Д\overset{˘}{s}\rangle \;\rangle \;\rangle }{dt}=\dot{\sigma }$ $+{\dot{S}}^{nf}$	$=\underset{D(t)}{\;\int \int \int \;}\left[\frac{\partial Д\overset{˘}{s}}{\partial t}+\nabla_{u}\cdot \left(Д\overset{˘}{s}\dot{u}\right)\right]dU$	[J K^−1^ s^−1^]	$Д\frac{d\overset{˘}{s}}{dt}$	$=\frac{\partial Д\overset{˘}{s}}{\partial t}+\nabla_{u}\cdot \left(Д\overset{˘}{s}\dot{u}\right)$	[J K^−1^ s^−1^ (m s^−1^)^−3^]
Probability	p(u|t) $=p_{u|t}$	$0=\frac{d\langle \;\langle \;\langle p_{u|t}\rangle \;\rangle \;\rangle }{dt}$	$=\underset{D(t)}{\;\int \int \int \;}\left[\frac{\partial p_{u|t}}{\partial t}+\nabla_{u}\cdot \left(p_{u|t}\dot{u}\right)\right]dU$	[s^−1^]	0	$=\frac{\partial p_{u|t}}{\partial t}+\nabla_{u}\cdot \left(p_{u|t}\dot{u}\right)$	[s^−1^ (m s^−1^)^−3^]

**Table 4 entropy-24-01493-t004:** Conservation Laws for the Velocivolumetric-Temporal Formulation (based on the Velocivolumetric Fluid Density ζ(u,x,t)).

Conserved Quantity	Density	Integral Equation			Differential Equation		
	φ(u,x,t) $=\zeta \underline{\overset{˘}{\varphi }}$	LHS = $\frac{dQ}{dt}$ = $\frac{DQ}{Dt}$	= RHS	SI Units	LHS	= RHS	SI Units
Fluid mass	ζ	$0=\dot{M}$ $=\frac{d\;[\;[\;[\langle \;\langle \;\langle \zeta \rangle \;\rangle \;\rangle ]\;]\;]}{dt}$	$=\left\{\begin{array}{c} \underset{\Omega (t)}{\;\int \int \int \;}\underset{D(x,t)}{\;\int \int \int \;}[\frac{\partial \zeta }{\partial t}+\nabla_{u}\cdot \left(\zeta \dot{u}\right)\\ +\nabla_{x}\cdot \left(\zeta u\right)]dUdV \end{array}\right.$	[kg s^−1^]	0	$=\frac{\partial \zeta }{\partial t}+\nabla_{u}\cdot \left(\zeta \dot{u}\right)$ $+\nabla_{x}\cdot \left(\zeta u\right)$	[kg s^−1^ m^−3^ (m s^−1^)^−3^]
Species mass	$\zeta {\underline{\overset{˘}{\chi }}}_{c}$ = $\zeta_{c}$	$\left({\dot{M}}_{c}\right)=\frac{d\;[\;\left[\;\left[\langle \;\langle \;\langle \zeta {\underline{\overset{˘}{\chi }}}_{c}\rangle \;\rangle \;\rangle \right]\;\right]\;]}{dt}$	$=\left\{\begin{array}{c} \underset{\Omega (t)}{\;\int \int \int \;}\underset{D(x,t)}{\;\int \int \int \;}[\frac{\partial \zeta {\underline{\overset{˘}{\chi }}}_{c}}{\partial t}+\nabla_{u}\cdot \left(\zeta {\underline{\overset{˘}{\chi }}}_{c}\dot{u}\right)\\ +\nabla_{x}\cdot \left(\zeta {\underline{\overset{˘}{\chi }}}_{c}u\right)]dUdV \end{array}\right.$	[kg_c_ s^−1^]	$\zeta \frac{d{\underline{\overset{˘}{\chi }}}_{c}}{dt}$	$=\left\{\begin{array}{c} \frac{\partial \zeta {\underline{\overset{˘}{\chi }}}_{c}}{\partial t}+\nabla_{u}\cdot \left(\zeta {\underline{\overset{˘}{\chi }}}_{c}\dot{u}\right)\\ +\nabla_{x}\cdot \left(\zeta {\underline{\overset{˘}{\chi }}}_{c}u\right) \end{array}\right.$	[kg_c_ s^−1^ m^−3^ (m s^−1^)^−3^]
Linear momentum	ζu	$\sum F=\frac{d\;[\;[\;[\langle \;\langle \;\langle \zeta u\rangle \;\rangle \;\rangle ]\;]\;]}{dt}$	$=\left\{\begin{array}{c} \underset{\Omega (t)}{\;\int \int \int \;}\underset{D(x,t)}{\;\int \int \int \;}[\frac{\partial \zeta u}{\partial t}+\nabla_{u}\cdot \left(\zeta u\dot{u}\right)\\ +\nabla_{x}\cdot \left(\zeta uu\right)]dUdV \end{array}\right.$	[(kg m s^−1^) s^−1^ = N]	$\zeta \frac{du}{dt}$	$=\left\{\begin{array}{c} \frac{\partial \zeta u}{\partial t}+\nabla_{u}\cdot \left(\zeta u\dot{u}\right)\\ +\nabla_{x}\cdot \left(\zeta uu\right) \end{array}\right.$	[N m^−3^ (m s^−1^)^−3^]
Angular momentum	$\zeta \left(\underline{\overset{˘}{r}}\times u\right)$	$\left.\begin{array}{c} \sum T\\ =\frac{d\;[\;\left[\;\left[\langle \;\langle \;\langle \zeta \left(\underline{\overset{˘}{r}}\times u\right)\rangle \;\rangle \;\rangle \right]\;\right]\;]}{dt} \end{array}\right\}$	$=\left\{\begin{array}{c} \underset{\Omega (t)}{\;\int \int \int \;}\underset{D(x,t)}{\;\int \int \int \;}[\frac{\partial \zeta \left(\underline{\overset{˘}{r}}\times u\right)}{\partial t}+\nabla_{u}\cdot \left(\zeta \left(\underline{\overset{˘}{r}}\times u\right)\dot{u}\right)\\ +\nabla_{x}\cdot \left(\zeta \left(\underline{\overset{˘}{r}}\times u\right)u\right)]dUdV \end{array}\right.$	[(kg m${}^{2}$ s^−1^) s^−1^ = N m]	$\zeta \frac{d\left(\underline{\overset{˘}{r}}\times u\right)}{dt}$	$=\left\{\begin{array}{c} \frac{\partial \zeta (\underline{\overset{˘}{r}}\times u)}{\partial t}+\nabla_{u}\cdot \left(\zeta \left(\underline{\overset{˘}{r}}\times u\right)\dot{u}\right)\\ +\nabla_{x}\cdot \left(\zeta \left(\underline{\overset{˘}{r}}\times u\right)u\right) \end{array}\right.$	[(N m) m^−3^ (m s^−1^)^−3^]
Energy	$\zeta \underline{\overset{˘}{e}}$	$\left.\begin{array}{c} \frac{DE}{Dt}=\frac{d\;[\;\left[\;\left[\langle \;\langle \;\langle \zeta \underline{\overset{˘}{e}}\rangle \;\rangle \;\rangle \right]\;\right]\;]}{dt}\\ =({\dot{Q}}_{in}+{\dot{W}}_{in}) \end{array}\right\}$	$=\left\{\begin{array}{c} \underset{\Omega (t)}{\;\int \int \int \;}\underset{D(x,t)}{\;\int \int \int \;}[\frac{\partial \zeta \underline{\overset{˘}{e}}}{\partial t}+\nabla_{u}\cdot \left(\zeta \underline{\overset{˘}{e}}\dot{u}\right)\\ +\nabla_{x}\cdot \left(\zeta \underline{\overset{˘}{e}}u\right)]dUdV \end{array}\right.$	[J s^−1^ = W]	$\zeta \frac{d\underline{\overset{˘}{e}}}{dt}$	$=\frac{\partial \zeta \underline{\overset{˘}{e}}}{\partial t}+\nabla_{u}\cdot \left(\zeta \underline{\overset{˘}{e}}\dot{u}\right)$ $+\nabla_{x}\cdot \left(\zeta \underline{\overset{˘}{e}}u\right)$	[J s^−1^ m^−3^ (m s^−1^)^−3^]
Charge (in solution)	$\zeta \underline{\overset{˘}{z}}$	$\left.\begin{array}{c} \frac{DZ}{Dt}=\frac{d\;[\;\left[\;\left[\langle \;\langle \;\langle \zeta \underline{\overset{˘}{z}}\rangle \;\rangle \;\rangle \right]\;\right]\;]}{dt}\\ =I+\left(\sum_{c}z_{c}{\dot{M}}_{c}\right) \end{array}\right\}$	$=\left\{\begin{array}{c} \underset{\Omega (t)}{\;\int \int \int \;}\underset{D(x,t)}{\;\int \int \int \;}[\frac{\partial \zeta \underline{\overset{˘}{z}}}{\partial t}+\nabla_{u}\cdot \left(\zeta \underline{\overset{˘}{z}}\dot{u}\right)\\ +\nabla_{x}\cdot \left(\zeta \underline{\overset{˘}{z}}u\right)]dUdV \end{array}\right.$	[C s^−1^ = A]	$\zeta \frac{d\underline{\overset{˘}{z}}}{dt}$	$=\frac{\partial \zeta \underline{\overset{˘}{z}}}{\partial t}+\nabla_{u}\cdot \left(\zeta \underline{\overset{˘}{z}}\dot{u}\right)$ $+\nabla_{x}\cdot \left(\zeta \underline{\overset{˘}{z}}u\right)$	[C s^−1^ m^−3^ (m s^−1^)^−3^]
Entropy	$\zeta \underline{\overset{˘}{s}}$	$\left.\begin{array}{c} \frac{DS}{Dt}=\frac{d\;[\;\left[\;\left[\langle \;\langle \;\langle \zeta \underline{\overset{˘}{s}}\rangle \;\rangle \;\rangle \right]\;\right]\;]}{dt}\\ =\dot{\sigma }+{\dot{S}}^{nf} \end{array}\right\}$	$=\left\{\begin{array}{c} \underset{\Omega (t)}{\;\int \int \int \;}\underset{D(x,t)}{\;\int \int \int \;}[\frac{\partial \zeta \underline{\overset{˘}{s}}}{\partial t}+\nabla_{u}\cdot \left(\zeta \underline{\overset{˘}{s}}\dot{u}\right)\\ +\nabla_{x}\cdot \left(\zeta \underline{\overset{˘}{s}}u\right)]dUdV \end{array}\right.$	[J K^−1^ s^−1^]	$\zeta \frac{d\underline{\overset{˘}{s}}}{dt}$	$=\left\{\begin{array}{c} \frac{\partial \zeta \underline{\overset{˘}{s}}}{\partial t}+\nabla_{u}\cdot \left(\zeta \underline{\overset{˘}{s}}\dot{u}\right)\\ +\nabla_{x}\cdot \left(\zeta \underline{\overset{˘}{s}}u\right) \end{array}\right.$	[J K^−1^ s^−1^ m^−3^ (m s^−1^)^−3^]
Probability	p(u,x|t) $=p_{u,x|t}$	$0=\frac{d\;[\;\left[\;\left[\langle \;\langle \;\langle p_{u,x|t}\rangle \;\rangle \;\rangle \right]\;\right]\;]}{dt}$	$=\left\{\begin{array}{c} \underset{\Omega (t)}{\;\int \int \int \;}\underset{D(x,t)}{\;\int \int \int \;}[\frac{\partial p_{u,x|t}}{\partial t}+\nabla_{u}\cdot \left(p_{u,x|t}\dot{u}\right)\\ +\nabla_{x}\cdot \left(p_{u,x|t}u\right)]dUdV \end{array}\right.$	[s^−1^]	0	$=\left\{\begin{array}{c} \frac{\partial p_{u,x|t}}{\partial t}+\nabla_{u}\cdot \left(p_{u,x|t}\dot{u}\right)\\ +\nabla_{x}\cdot \left(p_{u,x|t}u\right) \end{array}\right.$	[s^−1^ m^−3^ (m s^−1^)^−3^]

**Table 5 entropy-24-01493-t005:** Conservation Laws for the Velocimetric-Temporal Formulation (based on the Velocivolumetric Fluid Density ζ(u,x,t)).

Conserved Quantity	Density	Integral Equation			Differential Equation		
	φ(u,x,t) $=\zeta \underline{\overset{˘}{\varphi }}$	LHS = $\frac{\partial \alpha }{\partial t}=\frac{\partial (\rho \underline{\alpha })}{\partial t}$	= RHS	SI Units	LHS	= RHS	SI Units
Fluid mass	ζ	$\frac{\partial \rho }{\partial t}$ $=\frac{\partial \;\langle \;\langle \;\langle \zeta \rangle \;\rangle \;\rangle }{\partial t}$	$=\underset{D(x,t)}{\;\int \int \int \;}\left[\frac{\partial \zeta }{\partial t}+\nabla_{u}\cdot \left(\zeta \dot{u}\right)\right]dU$	[kg m^−3^ s^−1^]	$L_{\dot{u}∟t}^{\left(t\right)}\left(\zeta {\mathrm{vol}}_{u}^{3}\right)$	$=\left[\frac{\partial \zeta }{\partial t}+\nabla_{u}\cdot \left(\zeta \dot{u}\right)\right]dU$	[kg m^−3^ s^−1^]
Species mass	$\zeta {\underline{\overset{˘}{\chi }}}_{c}$ = $\zeta_{c}$	$\frac{\partial (\rho {\underline{\chi }}_{c})}{\partial t}=\frac{\partial \;\langle \;\langle \;\langle \zeta {\underline{\overset{˘}{\chi }}}_{c}\rangle \;\rangle \;\rangle }{\partial t}$	$=\underset{D(x,t)}{\;\int \int \int \;}\left[\frac{\partial \zeta {\underline{\overset{˘}{\chi }}}_{c}}{\partial t}+\nabla_{u}\cdot \left(\zeta {\underline{\overset{˘}{\chi }}}_{c}\dot{u}\right)\right]dU$	[kg_c_ m^−3^ s^−1^]	$L_{\dot{u}∟t}^{\left(t\right)}\left(\zeta {\underline{\overset{˘}{\chi }}}_{c}{\mathrm{vol}}_{u}^{3}\right)$	$=\left[\frac{\partial \zeta {\underline{\overset{˘}{\chi }}}_{c}}{\partial t}+\nabla_{u}\cdot \left(\zeta {\underline{\overset{˘}{\chi }}}_{c}\dot{u}\right)\right]dU$	[kg_c_ m^−3^ s^−1^]
Linear momentum	ζu	$\frac{\partial (\rho u)}{\partial t}=\frac{\partial \;\langle \;\langle \;\langle \zeta u\rangle \;\rangle \;\rangle }{\partial t}$	$=\underset{D(x,t)}{\;\int \int \int \;}\left[\frac{\partial \zeta u}{\partial t}+\nabla_{u}\cdot \left(\zeta u\dot{u}\right)\right]dU$	[(kg m s^−1^) m^−3^ s^−1^ = N m^−3^]	$L_{\dot{u}∟t}^{\left(t\right)}\left(\zeta u{\mathrm{vol}}_{u}^{3}\right)$	$=[\frac{\partial \zeta u}{\partial t}$ $+\nabla_{u}\cdot \left(\zeta u\dot{u}\right)]dU$	[N m^−3^]
Angular momentum	$\zeta \left(\underline{\overset{˘}{r}}\times u\right)$	$\left.\begin{array}{c} \frac{\partial (\rho \left(\underline{r}\times u\right))}{\partial t}\\ =\frac{\partial \;\langle \;\langle \;\langle \zeta \left(\underline{\overset{˘}{r}}\times u\right)\rangle \;\rangle \;\rangle }{\partial t} \end{array}\right\}$	$=\underset{D(x,t)}{\;\int \int \int \;}\left[\frac{\partial \zeta \left(\underline{\overset{˘}{r}}\times u\right)}{\partial t}+\nabla_{u}\cdot \left(\zeta \left(\underline{\overset{˘}{r}}\times u\right)\dot{u}\right)\right]dU$	[(kg m${}^{2}$ s^−1^) m^−3^ s^−1^ = (N m) m^−3^]	$\left.\begin{array}{c} L_{\dot{u}∟t}^{\left(t\right)}\left(\zeta \left(\underline{\overset{˘}{r}}\times u\right)\right)\\ {\mathrm{vol}}_{u}^{3}) \end{array}\right\}$	$=\left[\frac{\partial \zeta (\underline{\overset{˘}{r}}\times u)}{\partial t}+\nabla_{u}\cdot \left(\zeta \left(\underline{\overset{˘}{r}}\times u\right)\dot{u}\right)\right]dU$	[(N m) m^−3^]
Energy	$\zeta \underline{\overset{˘}{e}}$	$\frac{\partial (\rho \underline{e})}{\partial t}=\frac{\partial \;\langle \;\langle \;\langle \zeta \underline{\overset{˘}{e}}\rangle \;\rangle \;\rangle }{\partial t}$	$=\underset{D(x,t)}{\;\int \int \int \;}\left[\frac{\partial \zeta \underline{\overset{˘}{e}}}{\partial t}+\nabla_{u}\cdot \left(\zeta \underline{\overset{˘}{e}}\dot{u}\right)\right]dU$	[J m^−3^ s^−1^ = W m^−3^]	$L_{\dot{u}∟t}^{\left(t\right)}\left(\zeta \underline{\overset{˘}{e}}{\mathrm{vol}}_{u}^{3}\right)$	$=\left[\frac{\partial \zeta \underline{\overset{˘}{e}}}{\partial t}+\nabla_{u}\cdot \left(\zeta \underline{\overset{˘}{e}}\dot{u}\right)\right]dU$	[J m^−3^ s^−1^ = W m^−3^]
Charge (in solution)	$\zeta \underline{\overset{˘}{z}}$	$\frac{\partial (\rho \underline{z})}{\partial t}=\frac{\partial \;\langle \;\langle \;\langle \zeta \underline{\overset{˘}{z}}\rangle \;\rangle \;\rangle }{\partial t}$	$=\underset{D(x,t)}{\;\int \int \int \;}\left[\frac{\partial \zeta \underline{\overset{˘}{z}}}{\partial t}+\nabla_{u}\cdot \left(\zeta \underline{\overset{˘}{z}}\dot{u}\right)\right]dU$	[C m^−3^ s^−1^ = A m^−3^]	$L_{\dot{u}∟t}^{\left(t\right)}\left(\zeta \underline{\overset{˘}{z}}{\mathrm{vol}}_{u}^{3}\right)$	$=\left[\frac{\partial \zeta \underline{\overset{˘}{z}}}{\partial t}+\nabla_{u}\cdot \left(\zeta \underline{\overset{˘}{z}}\dot{u}\right)\right]dU$	[C m^−3^ s^−1^ = A m^−3^]
Entropy	$\zeta \underline{\overset{˘}{s}}$	$\frac{\partial (\rho \underline{s})}{\partial t}=\frac{\partial \;\langle \;\langle \;\langle \zeta \underline{\overset{˘}{s}}\rangle \;\rangle \;\rangle }{\partial t}$	$=\underset{D(x,t)}{\;\int \int \int \;}\left[\frac{\partial \zeta \underline{\overset{˘}{s}}}{\partial t}+\nabla_{u}\cdot \left(\zeta \underline{\overset{˘}{s}}\dot{u}\right)\right]dU$	[J K^−1^ m^−3^ s^−1^]	$L_{\dot{u}∟t}^{\left(t\right)}\left(\zeta \underline{\overset{˘}{s}}{\mathrm{vol}}_{u}^{3}\right)$	$=\left[\frac{\partial \zeta \underline{\overset{˘}{s}}}{\partial t}+\nabla_{u}\cdot \left(\zeta \underline{\overset{˘}{s}}\dot{u}\right)\right]dU$	[J K^−1^ m^−3^ s^−1^]
Probability	p(u,x|t) $=p_{u,x|t}$	$\frac{\partial p_{x|t}}{\partial t}=\frac{\partial \langle \;\langle \;\langle p_{u,x|t}\rangle \;\rangle \;\rangle }{\partial t}$	$=\underset{D(x,t)}{\;\int \int \int \;}\left[\frac{\partial p_{u,x|t}}{\partial t}+\nabla_{u}\cdot \left(p_{u,x|t}\dot{u}\right)\right]dU$	[m^−3^ s^−1^]	$L_{\dot{u}∟t}^{\left(t\right)}\left(p_{u,x|t}{\mathrm{vol}}_{u}^{3}\right)$	$=\left[\frac{\partial p_{u,x|t}}{\partial t}+\nabla_{u}\cdot \left(p_{u,x|t}\dot{u}\right)\right]dU$	[m^−3^ s^−1^]
	p(u|x,t) $=p_{u|x,t}$	$0=\frac{\partial \langle \;\langle \;\langle p_{u|x,t}\rangle \;\rangle \;\rangle }{\partial t}$	$=\underset{D(x,t)}{\;\int \int \int \;}\left[\frac{\partial p_{u|x,t}}{\partial t}+\nabla_{u}\cdot \left(p_{u|x,t}\dot{u}\right)\right]dU$	[s^−1^]	0	$=\frac{\partial p_{u|x,t}}{\partial t}+\nabla_{u}\cdot \left(p_{u|x,t}\dot{u}\right)$	[(m s^−1^)^−3^ s^−1^]

**Table 6 entropy-24-01493-t006:** Conservation Laws for the Volumetric-Temporal Formulation (based on the Velocivolumetric Fluid Density ζ(u,x,t)).

Conserved Quantity	Density	Integral Equation			Differential Equation		
	$\varphi (u,x,t)=\zeta \;\underline{\overset{˘}{\varphi }}$	LHS = $\frac{\partial \beta }{\partial t}=\frac{\partial (Д\overset{˘}{\beta })}{\partial t}$	= RHS	SI Units	LHS	= RHS	SI Units
Fluid mass	ζ	$\frac{\partial Д}{\partial t}$ $=\frac{\partial \;[\;[\;[\zeta ]\;]\;]}{\partial t}$	$=\underset{\Omega (u,t)}{\;\int \int \int \;}\left[\frac{\partial \zeta }{\partial t}+\nabla_{x}\cdot \left(\zeta u\right)\right]dV$	[kg (m s^−1^)^−3^ s^−1^]	$L_{u∟t}^{\left(t\right)}\left(\zeta {\mathrm{vol}}_{x}^{3}\right)$	$=\left[\frac{\partial \zeta }{\partial t}+\nabla_{x}\cdot \left(\zeta u\right)\right]dV$	[kg (m s^−1^)^−3^ s^−1^]
Species mass	$\zeta {\underline{\overset{˘}{\chi }}}_{c}$ = $\zeta_{c}$	$\frac{\partial (Д{\overset{˘}{\chi }}_{c})}{\partial t}=\frac{\partial \;[\;\left[\;\left[\zeta {\underline{\overset{˘}{\chi }}}_{c}\right]\;\right]\;]}{\partial t}$	$=\underset{\Omega (u,t)}{\;\int \int \int \;}\left[\frac{\partial \zeta {\underline{\overset{˘}{\chi }}}_{c}}{\partial t}+\nabla_{x}\cdot \left(\zeta {\underline{\overset{˘}{\chi }}}_{c}u\right)\right]dV$	[kg_c_ (m s^−1^)^−3^ s^−1^]	$L_{u∟t}^{\left(t\right)}\left(\zeta {\underline{\overset{˘}{\chi }}}_{c}{\mathrm{vol}}_{x}^{3}\right)$	$=\left[\frac{\partial \zeta {\underline{\overset{˘}{\chi }}}_{c}}{\partial t}+\nabla_{x}\cdot \left(\zeta {\underline{\overset{˘}{\chi }}}_{c}u\right)\right]dV$	[kg_c_ (m s^−1^)^−3^ s^−1^]
Linear momentum	ζu	$\frac{\partial (Дu)}{\partial t}=\frac{\partial \;[\;[\;[\zeta u]\;]\;]}{\partial t}$	$=\underset{\Omega (u,t)}{\;\int \int \int \;}\left[\frac{\partial \zeta u}{\partial t}+\nabla_{x}\cdot \left(\zeta uu\right)\right]dV$	[(kg m s^−1^) (m s^−1^)^−3^ s^−1^ = N (m s^−1^)^−3^]	$L_{u∟t}^{\left(t\right)}\left(\zeta u{\mathrm{vol}}_{x}^{3}\right)$	$=[\frac{\partial \zeta u}{\partial t}$ $+\nabla_{x}\cdot \left(\zeta uu\right)]dV$	[N (m s^−1^)^−3^]
Angular momentum	$\zeta \left(\underline{\overset{˘}{r}}\times u\right)$	$\left.\begin{array}{c} \frac{\partial (Д(\overset{˘}{r}\times u))}{\partial t}\\ =\frac{\partial \;[\;\left[\;\left[\zeta \left(\underline{\overset{˘}{r}}\times u\right)\right]\;\right]\;]}{\partial t} \end{array}\right\}$	$=\left\{\begin{array}{c} \underset{\Omega (u,t)}{\;\int \int \int \;}[\frac{\partial \zeta \left(\underline{\overset{˘}{r}}\times u\right)}{\partial t}\\ +\nabla_{x}\cdot \left(\zeta \left(\underline{\overset{˘}{r}}\times u\right)u\right)]dV \end{array}\right.$	[(kg m${}^{2}$ s^−1^) (m s^−1^)^−3^ s^−1^ = (N m) (m s^−1^)^−3^]	$\left.\begin{array}{c} L_{u∟t}^{\left(t\right)}\left(\zeta \left(\underline{\overset{˘}{r}}\times u\right)\right)\\ {\mathrm{vol}}_{x}^{3}) \end{array}\right\}$	$=\left\{\begin{array}{c} {}[\frac{\partial \zeta (\underline{\overset{˘}{r}}\times u)}{\partial t}\\ +\nabla_{x}\cdot \left(\zeta \left(\underline{\overset{˘}{r}}\times u\right)u\right)]dV \end{array}\right.$	[(N m) (m s^−1^)^−3^]
Energy	$\zeta \underline{\overset{˘}{e}}$	$\frac{\partial (Д\overset{˘}{e})}{\partial t}=\frac{\partial \;[\;\left[\;\left[\zeta \underline{\overset{˘}{e}}\right]\;\right]\;]}{\partial t}$	$=\underset{\Omega (u,t)}{\;\int \int \int \;}\left[\frac{\partial \zeta \underline{\overset{˘}{e}}}{\partial t}+\nabla_{x}\cdot \left(\zeta \underline{\overset{˘}{e}}u\right)\right]dV$	[J (m s^−1^)^−3^ s^−1^ = W (m s^−1^)^−3^]	$L_{u∟t}^{\left(t\right)}\left(\zeta \underline{\overset{˘}{e}}{\mathrm{vol}}_{x}^{3}\right)$	$=\left[\frac{\partial \zeta \underline{\overset{˘}{e}}}{\partial t}+\nabla_{x}\cdot \left(\zeta \underline{\overset{˘}{e}}u\right)\right]dV$	[J (m s^−1^)^−3^ s^−1^ = W (m s^−1^)^−3^]
Charge (in solution)	$\zeta \underline{\overset{˘}{z}}$	$\frac{\partial (Д\overset{˘}{z})}{\partial t}=\frac{\partial \;[\;\left[\;\left[\zeta \underline{\overset{˘}{z}}\right]\;\right]\;]}{\partial t}$	$=\underset{\Omega (u,t)}{\;\int \int \int \;}\left[\frac{\partial \zeta \underline{\overset{˘}{z}}}{\partial t}+\nabla_{x}\cdot \left(\zeta \underline{\overset{˘}{z}}u\right)\right]dV$	[C (m s^−1^)^−3^ s^−1^ = A (m s^−1^)^−3^]	$L_{u∟t}^{\left(t\right)}\left(\zeta \underline{\overset{˘}{z}}{\mathrm{vol}}_{x}^{3}\right)$	$=\left[\frac{\partial \zeta \underline{\overset{˘}{z}}}{\partial t}+\nabla_{x}\cdot \left(\zeta \underline{\overset{˘}{z}}u\right)\right]dV$	[C (m s^−1^)^−3^ s^−1^ = A (m s^−1^)^−3^]
Entropy	$\zeta \underline{\overset{˘}{s}}$	$\frac{\partial (Д\overset{˘}{s})}{\partial t}=\frac{\partial \;[\;\left[\;\left[\zeta \underline{\overset{˘}{s}}\right]\;\right]\;]}{\partial t}$	$=\underset{\Omega (u,t)}{\;\int \int \int \;}\left[\frac{\partial \zeta \underline{\overset{˘}{s}}}{\partial t}+\nabla_{x}\cdot \left(\zeta \underline{\overset{˘}{s}}u\right)\right]dV$	[J K^−1^ (m s^−1^)^−3^ s^−1^]	$L_{u∟t}^{\left(t\right)}\left(\zeta \underline{\overset{˘}{s}}{\mathrm{vol}}_{x}^{3}\right)$	$=\left[\frac{\partial \zeta \underline{\overset{˘}{s}}}{\partial t}+\nabla_{x}\cdot \left(\zeta \underline{\overset{˘}{s}}u\right)\right]dV$	[J K^−1^ (m s^−1^)^−3^ s^−1^]
Probability	p(u,x|t) $=p_{u,x|t}$	$\frac{\partial p_{u|t}}{\partial t}=\frac{\partial [\;\left[\;\left[p_{u,x|t}\right]\;\right]\;]}{\partial t}$	$=\underset{\Omega (u,t)}{\;\int \int \int \;}\left[\frac{\partial p_{u,x|t}}{\partial t}+\nabla_{x}\cdot \left(p_{u,x|t}u\right)\right]dV$	[(m s^−1^)^−3^ s^−1^]	$L_{u∟t}^{\left(t\right)}\left(p_{u,x|t}{\mathrm{vol}}_{x}^{3}\right)$	$=\left[\frac{\partial p_{u,x|t}}{\partial t}+\nabla_{x}\cdot \left(p_{u,x|t}u\right)\right]dV$	[(m s^−1^)^−3^ s^−1^]
	p(x|u,t) $=p_{x|u,t}$	$0=\frac{\partial [\;\left[\;\left[p_{x|u,t}\right]\;\right]\;]}{\partial t}$	$=\underset{\Omega (u,t)}{\;\int \int \int \;}\left[\frac{\partial p_{x|u,t}}{\partial t}+\nabla_{x}\cdot \left(p_{x|u,t}u\right)\right]dV$	[s^−1^]	0	$=\frac{\partial p_{x|u,t}}{\partial t}+\nabla_{x}\cdot \left(p_{x|u,t}u\right)$	[m^−3^ s^−1^]

**Table 7 entropy-24-01493-t007:** Conservation Laws for the Velocimetric-Spatial (Time-Independent) Formulation (based on the Velocivolumetric Fluid Density ζ(u,x)).

Conserved Quantity	Density	Integral Equation			Differential Equation		
	φ(u,x) $=\zeta \overset{˘}{\underline{\varphi }}$	LHS = $\nabla_{x}\alpha =\nabla_{x}\left(\rho \underline{\alpha }\right)$	= RHS	SI Units	LHS	= RHS	SI Units
Fluid mass	ζ	$\nabla_{x}\rho$ $=\nabla_{x}\langle \;\langle \;\langle \zeta \rangle \;\rangle \;\rangle$	$=\underset{D(x)}{\;\int \int \int \;}\left(\nabla_{x}\zeta +\nabla_{u}\cdot \left(\zeta \;G^{⊤}\right)\right)dU$	[kg m^−4^ = kg m^−3^ m^−1^]	$L_{G^{⊤}∟x}^{\left(x\right)}\left(\zeta {\mathrm{vol}}_{u}^{3}\right)$	$=\left(\nabla_{x}\zeta +\nabla_{u}\cdot \left(\zeta \;G^{⊤}\right)\right)dU$	[kg m^−4^ = kg m^−3^ m^−1^]
Species mass	$\zeta {\overset{˘}{\underline{\chi }}}_{c}$ $=\zeta_{c}$	$\left.\begin{array}{c} \nabla_{x}\left(\rho {\underline{\chi }}_{c}\right)=\nabla_{x}\left(\rho_{c}\right)\\ =\nabla_{x}\langle \;\langle \;\langle \zeta {\overset{˘}{\underline{\chi }}}_{c}\rangle \;\rangle \;\rangle \end{array}\right\}$	$=\underset{D(x)}{\;\int \int \int \;}\left(\nabla_{x}\left(\zeta {\overset{˘}{\underline{\chi }}}_{c}\right)+\nabla_{u}\cdot \left(\zeta {\overset{˘}{\underline{\chi }}}_{c}\;G^{⊤}\right)\right)dU$	[kg_c_ m^−4^ = kg_c_ m^−3^ m^−1^]	$L_{G^{⊤}∟x}^{\left(x\right)}\left(\zeta {\overset{˘}{\underline{\chi }}}_{c}{\mathrm{vol}}_{u}^{3}\right)$	$=\left(\nabla_{x}\left(\zeta {\overset{˘}{\underline{\chi }}}_{c}\right)+\nabla_{u}\cdot \left(\zeta {\overset{˘}{\underline{\chi }}}_{c}\;G^{⊤}\right)\right)dU$	[kg_c_ m^−4^ = kg_c_ m^−3^ m^−1^]
Linear momentum	ζu	$\nabla_{x}\left(\rho u\right)$ $=\nabla_{x}\langle \;\langle \;\langle \zeta u\rangle \;\rangle \;\rangle$	$=\underset{D(x)}{\;\int \int \int \;}\left(\nabla_{x}\left(\zeta u\right)+\nabla_{u}\cdot \left(\zeta u\;G^{⊤}\right)\right)dU$	[kg m^−3^ s^−1^ = (kg m s^−1^) m^−3^ m^−1^]	$L_{G^{⊤}∟x}^{\left(x\right)}\left(\zeta u{\mathrm{vol}}_{u}^{3}\right)$	$=\left(\nabla_{x}\left(\zeta u\right)+\nabla_{u}\cdot \left(\zeta u\;G^{⊤}\right)\right)dU$	[kg m^−3^ s^−1^]
Angular momentum	$\zeta \left(\underline{\overset{˘}{r}}\times u\right)$	$\left.\begin{array}{c} \nabla_{x}\left(\rho \left(\underline{r}\times u\right)\right)\\ =\nabla_{x}\langle \;\langle \;\langle \zeta \left(\underline{\overset{˘}{r}}\times u\right)\rangle \;\rangle \;\rangle \end{array}\right\}$	$=\left\{\begin{array}{c} \underset{D(x)}{\;\int \int \int \;}(\nabla_{x}\left(\zeta \left(\underline{\overset{˘}{r}}\times u\right)\right)\\ +\nabla_{u}\cdot \left(\zeta \left(\underline{\overset{˘}{r}}\times u\right)\;G^{⊤}\right))dU \end{array}\right.$	[kg m^−2^ s^−1^ = (kg m^2^ s^−1^) m^−3^ m^−1^]	$\left.\begin{array}{c} L_{G^{⊤}∟x}^{\left(x\right)}(\zeta \left(\underline{\overset{˘}{r}}\times u\right)\\ {\mathrm{vol}}_{u}^{3}) \end{array}\right\}$	$=\left\{\begin{array}{c} (\nabla_{x}\left(\zeta \left(\underline{\overset{˘}{r}}\times u\right)\right)\\ +\nabla_{u}\cdot \left(\zeta \left(\underline{\overset{˘}{r}}\times u\right)\;G^{⊤}\right))dU \end{array}\right.$	[kg m^−2^ s^−1^]
Energy	$\zeta \overset{˘}{\underline{e}}$	$\nabla_{x}\left(\rho \underline{e}\right)$ $=\nabla_{x}\langle \;\langle \;\langle \zeta \overset{˘}{\underline{e}}\rangle \;\rangle \;\rangle$	$=\underset{D(x)}{\;\int \int \int \;}\left(\nabla_{x}\left(\zeta \overset{˘}{\underline{e}}\right)+\nabla_{u}\cdot \left(\zeta \overset{˘}{\underline{e}}\;G^{⊤}\right)\right)dU$	[J m^−3^ m^−1^]	$L_{G^{⊤}∟x}^{\left(x\right)}\left(\zeta \overset{˘}{\underline{e}}{\mathrm{vol}}_{u}^{3}\right)$	$=\left(\nabla_{x}\left(\zeta \overset{˘}{\underline{e}}\right)+\nabla_{u}\cdot \left(\zeta \overset{˘}{\underline{e}}\;G^{⊤}\right)\right)dU$	[J m^−3^ m^−1^]
Charge (in solution)	$\zeta \overset{˘}{\underline{z}}$	$\nabla_{x}\left(\rho \underline{z}\right)$ $=\nabla_{x}\langle \;\langle \;\langle \zeta \overset{˘}{\underline{z}}\rangle \;\rangle \;\rangle$	$=\underset{D(x)}{\;\int \int \int \;}\left(\nabla_{x}\left(\zeta \overset{˘}{\underline{z}}\right)+\nabla_{u}\cdot \left(\zeta \overset{˘}{\underline{z}}\;G^{⊤}\right)\right)dU$	[C m^−3^ m^−1^]	$L_{G^{⊤}∟x}^{\left(x\right)}\left(\zeta \overset{˘}{\underline{z}}{\mathrm{vol}}_{u}^{3}\right)$	$=\left(\nabla_{x}\left(\zeta \overset{˘}{\underline{z}}\right)+\nabla_{u}\cdot \left(\zeta \overset{˘}{\underline{z}}\;G^{⊤}\right)\right)dU$	[C m^−3^ m^−1^]
Entropy	$\zeta \overset{˘}{\underline{s}}$	$\nabla_{x}\left(\rho \underline{s}\right)$ $=\nabla_{x}\langle \;\langle \;\langle \zeta \overset{˘}{\underline{s}}\rangle \;\rangle \;\rangle$	$=\underset{D(x)}{\;\int \int \int \;}\left(\nabla_{x}\left(\zeta \overset{˘}{\underline{s}}\right)+\nabla_{u}\cdot \left(\zeta \overset{˘}{\underline{s}}\;G^{⊤}\right)\right)dU$	[J K^−1^ m^−3^ m^−1^]	$L_{G^{⊤}∟x}^{\left(x\right)}\left(\zeta \overset{˘}{\underline{s}}{\mathrm{vol}}_{u}^{3}\right)$	$=\left(\nabla_{x}\left(\zeta \overset{˘}{\underline{s}}\right)+\nabla_{u}\cdot \left(\zeta \overset{˘}{\underline{s}}\;G^{⊤}\right)\right)dU$	[J K^−1^ m^−3^ m^−1^]
Probability	p(u,x) $=p_{u,x}$	$\nabla_{x}p_{x}$ $=\nabla_{x}\langle \;\langle \;\langle p_{u,x}\rangle \;\rangle \;\rangle$	$=\underset{D(x)}{\;\int \int \int \;}\left(\nabla_{x}p_{u,x}+\nabla_{u}\cdot \left(p_{u,x}\;G^{⊤}\right)\right)dU$	[m^−3^ m^−1^]	$L_{G^{⊤}∟x}^{\left(x\right)}\left(p_{u,x}{\mathrm{vol}}_{u}^{3}\right)$	$=\left(\nabla_{x}p_{u,x}+\nabla_{u}\cdot \left(p_{u,x}\;G^{⊤}\right)\right)dU$	[m^−3^ m^−1^]
	p(u|x) $=p_{u|x}$	$0=\nabla_{x}\langle \;\langle \;\langle p_{u|x}\rangle \;\rangle \;\rangle$	$=\underset{D(x)}{\;\int \int \int \;}\left(\nabla_{x}p_{u|x}+\nabla_{u}\cdot \left(p_{u|x}\;G^{⊤}\right)\right)dU$	[m^−1^]	0	$=\nabla_{x}p_{u|x}+\nabla_{u}\cdot \left(p_{u|x}\;G^{⊤}\right)$	[(m s^−1^)^−3^ m^−1^]

**Table 8 entropy-24-01493-t008:** Conservation Laws for the Volumetric-Velocital (Time-Independent) Formulation (based on the Velocivolumetric Fluid Density ζ(u,x)).

Conserved Quantity	Density	Integral Equation			Differential Equation		
	φ(u,x) $=\zeta \;\overset{˘}{\underline{\varphi }}$	LHS = $\nabla_{u}\beta$ $=\nabla_{u}\left(Д\overset{˘}{\beta }\right)$	= RHS	SI Units	LHS	= RHS	SI Units
Fluid mass	ζ	$\nabla_{u}Д$ $=\nabla_{u}\left[\;\left[\;\left[\zeta \right]\;\right]\;\right]$	$=\underset{\Omega (u)}{\;\int \int \int \;}\left(\nabla_{u}\zeta +\nabla_{x}\cdot \left(\zeta \;\Gamma^{⊤}\right)\right)dV$	[kg (m s^−1^)^−4^ = kg (m s^−1^)^−3^ (m s^−1^)^−1^]	$L_{\Gamma^{⊤}∟u}^{\left(u\right)}\left(\zeta {\mathrm{vol}}_{x}^{3}\right)$	$=\left(\nabla_{u}\zeta +\nabla_{x}\cdot \left(\zeta \;\Gamma^{⊤}\right)\right)dV$	[kg (m s^−1^)^−4^ = kg (m s^−1^)^−3^ (m s^−1^)^−1^]
Species mass	$\zeta {\overset{˘}{\underline{\chi }}}_{c}$ $=\zeta_{c}$	$\left.\begin{array}{c} \nabla_{u}\left(Д{\overset{˘}{\chi }}_{c}\right)=\nabla_{u}\left({Д}_{c}\right)\\ =\nabla_{u}\left[\;\left[\;\left[\zeta {\overset{˘}{\underline{\chi }}}_{c}\right]\;\right]\;\right] \end{array}\right\}$	$=\underset{\Omega (u)}{\;\int \int \int \;}\left(\nabla_{u}\left(\zeta {\overset{˘}{\underline{\chi }}}_{c}\right)+\nabla_{x}\cdot \left(\zeta {\overset{˘}{\underline{\chi }}}_{c}\;\Gamma^{⊤}\right)\right)dV$	[kg_c_ (m s^−1^)^−4^ = kg_c_ (m s^−1^)^−3^ (m s^−1^)^−1^]	$L_{\Gamma^{⊤}∟u}^{\left(u\right)}\left(\zeta {\overset{˘}{\underline{\chi }}}_{c}{\mathrm{vol}}_{x}^{3}\right)$	$=\left(\nabla_{u}\left(\zeta {\overset{˘}{\underline{\chi }}}_{c}\right)+\nabla_{x}\cdot \left(\zeta {\overset{˘}{\underline{\chi }}}_{c}\;\Gamma^{⊤}\right)\right)dV$	[kg_c_ (m s^−1^)^−4^ = kg_c_ (m s^−1^)^−3^ (m s^−1^)^−1^]
Linear momentum	ζu	$\nabla_{u}\left(Дu\right)$ $=\nabla_{u}\left[\;\left[\;\left[\zeta u\right]\;\right]\;\right]$	$=\underset{\Omega (u)}{\;\int \int \int \;}\left(\nabla_{u}\left(\zeta u\right)+\nabla_{x}\cdot \left(\zeta u\;\Gamma^{⊤}\right)\right)dV$	[kg (m s^−1^)^−3^ = (kg m s^−1^) (m s^−1^)^−3^ (m s^−1^)^−1^]	$L_{\Gamma^{⊤}∟u}^{\left(u\right)}\left(\zeta u{\mathrm{vol}}_{x}^{3}\right)$	$=\left(\nabla_{u}\left(\zeta u\right)+\nabla_{x}\cdot \left(\zeta u\;\Gamma^{⊤}\right)\right)dV$	[kg (m s^−1^)^−3^]
Angular momentum	$\zeta \left(\underline{\overset{˘}{r}}\times u\right)$	$\left.\begin{array}{c} \nabla_{u}\left(Д(\overset{˘}{r}\times u)\right)\\ =\nabla_{u}\left[\;\left[\;\left[\zeta \left(\underline{\overset{˘}{r}}\times u\right)\right]\;\right]\;\right] \end{array}\right\}$	$=\left\{\begin{array}{c} \underset{\Omega (u)}{\;\int \int \int \;}(\nabla_{u}\left(\zeta \left(\underline{\overset{˘}{r}}\times u\right)\right)\\ +\nabla_{x}\cdot \left(\zeta \left(\underline{\overset{˘}{r}}\times u\right)\;\Gamma^{⊤}\right))dV \end{array}\right.$	[kg m (m s^−1^)^−3^ = (kg m^2^ s^−1^) (m s^−1^)^−3^ (m s^−1^)^−1^]	$\left.\begin{array}{c} L_{\Gamma^{⊤}∟u}^{\left(u\right)}(\zeta \left(\underline{\overset{˘}{r}}\times u\right)\\ {\mathrm{vol}}_{x}^{3}) \end{array}\right\}$	$=\left\{\begin{array}{c} (\nabla_{u}\left(\zeta \left(\underline{\overset{˘}{r}}\times u\right)\right)\\ +\nabla_{x}\cdot \left(\zeta \left(\underline{\overset{˘}{r}}\times u\right)\;\Gamma^{⊤}\right))dV \end{array}\right.$	[kg m (m s^−1^)^−3^]
Energy	$\zeta \overset{˘}{\underline{e}}$	$\nabla_{u}\left(Д\overset{˘}{e}\right)$ $=\nabla_{u}\left[\;\left[\;\left[\zeta \overset{˘}{\underline{e}}\right]\;\right]\;\right]$	$=\underset{\Omega (u)}{\;\int \int \int \;}\left(\nabla_{u}\left(\zeta \overset{˘}{\underline{e}}\right)+\nabla_{x}\cdot \left(\zeta \overset{˘}{\underline{e}}\;\Gamma^{⊤}\right)\right)dV$	[J (m s^−1^)^−3^ (m s^−1^)^−1^]	$L_{\Gamma^{⊤}∟u}^{\left(u\right)}\left(\zeta \overset{˘}{\underline{e}}{\mathrm{vol}}_{x}^{3}\right)$	$=\left(\nabla_{u}\left(\zeta \overset{˘}{\underline{e}}\right)+\nabla_{x}\cdot \left(\zeta \overset{˘}{\underline{e}}\;\Gamma^{⊤}\right)\right)dV$	[J (m s^−1^)^−3^ (m s^−1^)^−1^]
Charge (in solution)	$\zeta \overset{˘}{\underline{z}}$	$\nabla_{u}\left(Д\overset{˘}{z}\right)$ $=\nabla_{u}\left[\;\left[\;\left[\zeta \overset{˘}{\underline{z}}\right]\;\right]\;\right]$	$=\underset{\Omega (u)}{\;\int \int \int \;}\left(\nabla_{u}\left(\zeta \overset{˘}{\underline{z}}\right)+\nabla_{x}\cdot \left(\zeta \overset{˘}{\underline{z}}\;\Gamma^{⊤}\right)\right)dV$	[C (m s^−1^)^−3^ (m s^−1^)^−1^]	$L_{\Gamma^{⊤}∟u}^{\left(u\right)}\left(\zeta \overset{˘}{\underline{z}}{\mathrm{vol}}_{x}^{3}\right)$	$=\left(\nabla_{u}\left(\zeta \overset{˘}{\underline{z}}\right)+\nabla_{x}\cdot \left(\zeta \overset{˘}{\underline{z}}\;\Gamma^{⊤}\right)\right)dV$	[C (m s^−1^)^−3^ (m s^−1^)^−1^]
Entropy	$\zeta \overset{˘}{\underline{s}}$	$\nabla_{u}\left(Д\overset{˘}{s}\right)$ $=\nabla_{u}\left[\;\left[\;\left[\zeta \overset{˘}{\underline{s}}\right]\;\right]\;\right]$	$=\underset{\Omega (u)}{\;\int \int \int \;}\left(\nabla_{u}\left(\zeta \overset{˘}{\underline{s}}\right)+\nabla_{x}\cdot \left(\zeta \overset{˘}{\underline{s}}\;\Gamma^{⊤}\right)\right)dV$	[J K^−1^ (m s^−1^)^−3^ (m s^−1^)^−1^]	$L_{\Gamma^{⊤}∟u}^{\left(u\right)}\left(\zeta \overset{˘}{\underline{s}}{\mathrm{vol}}_{x}^{3}\right)$	$=\left(\nabla_{u}\left(\zeta \overset{˘}{\underline{s}}\right)+\nabla_{x}\cdot \left(\zeta \overset{˘}{\underline{s}}\;\Gamma^{⊤}\right)\right)dV$	[J K^−1^ (m s^−1^)^−3^ (m s^−1^)^−1^]
Probability	p(u,x) $=p_{u,x}$	$\nabla_{u}p_{u}$ $=\nabla_{u}\left[\;\left[\;\left[p_{u,x}\right]\;\right]\;\right]$	$=\underset{\Omega (u)}{\;\int \int \int \;}\left(\nabla_{u}p_{u,x}+\nabla_{x}\cdot \left(p_{u,x}\;\Gamma^{⊤}\right)\right)dV$	[(m s^−1^)^−3^ (m s^−1^)^−1^]	$L_{\Gamma^{⊤}∟u}^{\left(u\right)}\left(p_{u,x}{\mathrm{vol}}_{x}^{3}\right)$	$=\left(\nabla_{u}p_{u,x}+\nabla_{x}\cdot \left(p_{u,x}\;\Gamma^{⊤}\right)\right)dV$	[(m s^−1^)^−3^ (m s^−1^)^−1^]
	p(x|u) $=p_{x|u}$	$0=\nabla_{u}\left[\;\left[\;\left[p_{x|u}\right]\;\right]\;\right]$	$=\underset{\Omega (u)}{\;\int \int \int \;}\left(\nabla_{u}p_{x|u}+\nabla_{x}\cdot \left(p_{x|u}\;\Gamma^{⊤}\right)\right)dV$	[(m s^−1^)^−1^]	0	$=\nabla_{u}p_{x|u}+\nabla_{x}\cdot \left(p_{x|u}\;\Gamma^{⊤}\right)$	[m^−3^ (m s^−1^)^−1^]

**Table 9 entropy-24-01493-t009:** Conservation Laws for the Velocimetric-Spatiotemporal Formulation (based on the Velocivolumetric Fluid Density ζ(u,x,t)).

Conserved Quantity	Density	Integral Equation			Differential Equation		
	φ(u,x,t) $=\zeta \;\overset{˘}{\underline{\varphi }}$	LHS = ${▾}_{x}\alpha$ $={▾}_{x}\left(\rho \underline{\alpha }\right)$	= RHS	SI Units	LHS	= RHS	SI Units
Fluid mass	ζ	${▾}_{x}\rho$ $={▾}_{x}\langle \;\langle \;\langle \zeta \rangle \;\rangle \;\rangle$	$=\underset{D(x,t)}{\;\int \int \int \;}\left({▾}_{x}\zeta +\nabla_{u}\cdot \left(\zeta \;{\tilde{G}}^{⊤}\right)\right)dU$	[kg m^−3^ $d_{x,t}^{-1}$]	$L_{{\tilde{G}}^{⊤}∟x,t}^{\left(x,t\right)}\left(\zeta {\mathrm{vol}}_{u}^{3}\right)$	$=\left({▾}_{x}\zeta +\nabla_{u}\cdot \left(\zeta \;{\tilde{G}}^{⊤}\right)\right)dU$	[kg m^−3^ $d_{x,t}^{-1}$]
Species mass	$\zeta {\overset{˘}{\underline{\chi }}}_{c}$ $=\zeta_{c}$	$\left.\begin{array}{c} {▾}_{x}\left(\rho {\underline{\chi }}_{c}\right)={▾}_{x}\left(\rho_{c}\right)\\ ={▾}_{x}\langle \;\langle \;\langle \zeta {\overset{˘}{\underline{\chi }}}_{c}\rangle \;\rangle \;\rangle \end{array}\right\}$	$=\underset{D(x,t)}{\;\int \int \int \;}\left({▾}_{x}\left(\zeta {\overset{˘}{\underline{\chi }}}_{c}\right)+\nabla_{u}\cdot \left(\zeta {\overset{˘}{\underline{\chi }}}_{c}\;{\tilde{G}}^{⊤}\right)\right)dU$	[kg_c_ m^−3^ $d_{x,t}^{-1}$]	$L_{{\tilde{G}}^{⊤}∟x,t}^{\left(x,t\right)}\left(\zeta {\overset{˘}{\underline{\chi }}}_{c}{\mathrm{vol}}_{u}^{3}\right)$	$=\left({▾}_{x}\left(\zeta {\overset{˘}{\underline{\chi }}}_{c}\right)+\nabla_{u}\cdot \left(\zeta {\overset{˘}{\underline{\chi }}}_{c}\;{\tilde{G}}^{⊤}\right)\right)dU$	[kg_c_ m^−3^ $d_{x,t}^{-1}$]
Linear momentum	ζu	${▾}_{x}\left(\rho u\right)$ $={▾}_{x}\langle \;\langle \;\langle \zeta u\rangle \;\rangle \;\rangle$	$=\underset{D(x,t)}{\;\int \int \int \;}\left({▾}_{x}\left(\zeta u\right)+\nabla_{u}\cdot \left(\zeta u\;{\tilde{G}}^{⊤}\right)\right)dU$	[(kg m s^−1^) m^−3^ $d_{x,t}^{-1}$]	$L_{{\tilde{G}}^{⊤}∟x,t}^{\left(x,t\right)}\left(\zeta u{\mathrm{vol}}_{u}^{3}\right)$	$=\left({▾}_{x}\left(\zeta u\right)+\nabla_{u}\cdot \left(\zeta u\;{\tilde{G}}^{⊤}\right)\right)dU$	[(kg m s^−1^) m^−3^ $d_{x,t}^{-1}$]
Angular momentum	$\zeta \left(\underline{\overset{˘}{r}}\times u\right)$	$\left.\begin{array}{c} {▾}_{x}\left(\rho \left(\underline{r}\times u\right)\right)\\ ={▾}_{x}\langle \;\langle \;\langle \zeta \left(\underline{\overset{˘}{r}}\times u\right)\rangle \;\rangle \;\rangle \end{array}\right\}$	$=\left\{\begin{array}{c} \underset{D(x,t)}{\;\int \int \int \;}({▾}_{x}\left(\zeta \left(\underline{\overset{˘}{r}}\times u\right)\right)\\ +\nabla_{u}\cdot \left(\zeta \left(\underline{\overset{˘}{r}}\times u\right)\;{\tilde{G}}^{⊤}\right))dU \end{array}\right.$	[(kg m^2^ s^−1^) m^−3^ $d_{x,t}^{-1}$]	$\left.\begin{array}{c} L_{{\tilde{G}}^{⊤}∟x,t}^{\left(x,t\right)}(\zeta \left(\underline{\overset{˘}{r}}\times u\right)\\ {\mathrm{vol}}_{u}^{3}) \end{array}\right\}$	$=\left\{\begin{array}{c} ({▾}_{x}\left(\zeta \left(\underline{\overset{˘}{r}}\times u\right)\right)\\ +\nabla_{u}\cdot \left(\zeta \left(\underline{\overset{˘}{r}}\times u\right)\;{\tilde{G}}^{⊤}\right))dU \end{array}\right.$	[(kg m^2^ s^−1^) m^−3^ $d_{x,t}^{-1}$]
Energy	$\zeta \overset{˘}{\underline{e}}$	${▾}_{x}\left(\rho \underline{e}\right)$ $={▾}_{x}\langle \;\langle \;\langle \zeta \overset{˘}{\underline{e}}\rangle \;\rangle \;\rangle$	$=\underset{D(x,t)}{\;\int \int \int \;}\left({▾}_{x}\left(\zeta \overset{˘}{\underline{e}}\right)+\nabla_{u}\cdot \left(\zeta \overset{˘}{\underline{e}}\;{\tilde{G}}^{⊤}\right)\right)dU$	[J m^−3^ $d_{x,t}^{-1}$]	$L_{{\tilde{G}}^{⊤}∟x,t}^{\left(x,t\right)}\left(\zeta \overset{˘}{\underline{e}}{\mathrm{vol}}_{u}^{3}\right)$	$=\left({▾}_{x}\left(\zeta \overset{˘}{\underline{e}}\right)+\nabla_{u}\cdot \left(\zeta \overset{˘}{\underline{e}}\;{\tilde{G}}^{⊤}\right)\right)dU$	[J m^−3^ $d_{x,t}^{-1}$]
Charge (in solution)	$\zeta \overset{˘}{\underline{z}}$	${▾}_{x}\left(\rho \underline{z}\right)$ $={▾}_{x}\langle \;\langle \;\langle \zeta \overset{˘}{\underline{z}}\rangle \;\rangle \;\rangle$	$=\underset{D(x,t)}{\;\int \int \int \;}\left({▾}_{x}\left(\zeta \overset{˘}{\underline{z}}\right)+\nabla_{u}\cdot \left(\zeta \overset{˘}{\underline{z}}\;{\tilde{G}}^{⊤}\right)\right)dU$	[C m^−3^ $d_{x,t}^{-1}$]	$L_{{\tilde{G}}^{⊤}∟x,t}^{\left(x,t\right)}\left(\zeta \overset{˘}{\underline{z}}{\mathrm{vol}}_{u}^{3}\right)$	$=\left({▾}_{x}\left(\zeta \overset{˘}{\underline{z}}\right)+\nabla_{u}\cdot \left(\zeta \overset{˘}{\underline{z}}\;{\tilde{G}}^{⊤}\right)\right)dU$	[C m^−3^ $d_{x,t}^{-1}$]
Entropy	$\zeta \overset{˘}{\underline{s}}$	${▾}_{x}\left(\rho \underline{s}\right)$ $={▾}_{x}\langle \;\langle \;\langle \zeta \overset{˘}{\underline{s}}\rangle \;\rangle \;\rangle$	$=\underset{D(x,t)}{\;\int \int \int \;}\left({▾}_{x}\left(\zeta \overset{˘}{\underline{s}}\right)+\nabla_{u}\cdot \left(\zeta \overset{˘}{\underline{s}}\;{\tilde{G}}^{⊤}\right)\right)dU$	[J K^−1^ m^−3^ $d_{x,t}^{-1}$]	$L_{{\tilde{G}}^{⊤}∟x,t}^{\left(x,t\right)}\left(\zeta \overset{˘}{\underline{s}}{\mathrm{vol}}_{u}^{3}\right)$	$=\left({▾}_{x}\left(\zeta \overset{˘}{\underline{s}}\right)+\nabla_{u}\cdot \left(\zeta \overset{˘}{\underline{s}}\;{\tilde{G}}^{⊤}\right)\right)dU$	[J K^−1^ m^−3^ $d_{x,t}^{-1}$]
Probability	p(u,x|t) $=p_{u,x|t}$	${▾}_{x}p_{x|t}$ $={▾}_{x}\langle \;\langle \;\langle p_{u,x|t}\rangle \;\rangle \;\rangle$	$=\underset{D(x,t)}{\;\int \int \int \;}\left({▾}_{x}p_{u,x|t}+\nabla_{u}\cdot \left(p_{u,x|t}\;{\tilde{G}}^{⊤}\right)\right)dU$	[m^−3^ $d_{x,t}^{-1}$]	$\left.\begin{array}{c} L_{{\tilde{G}}^{⊤}∟x,t}^{\left(x,t\right)}(p_{u,x|t}\\ {\mathrm{vol}}_{u}^{3}) \end{array}\right\}$	$=\left\{\begin{array}{c} ({▾}_{x}p_{u,x|t}\\ +\nabla_{u}\cdot \left(p_{u,x|t}\;{\tilde{G}}^{⊤}\right))dU \end{array}\right.$	[m^−3^ $d_{x,t}^{-1}$]
	p(u|x,t) $=p_{u|x,t}$	$0={▾}_{x}\langle \;\langle \;\langle p_{u|x,t}\rangle \;\rangle \;\rangle$	$=\underset{D(x,t)}{\;\int \int \int \;}\left({▾}_{x}p_{u|x,t}+\nabla_{u}\cdot \left(p_{u|x,t}\;{\tilde{G}}^{⊤}\right)\right)dU$	[$d_{x,t}^{-1}$]	0	$={▾}_{x}p_{u|x,t}+\nabla_{u}\cdot \left(p_{u|x,t}\;{\tilde{G}}^{⊤}\right)$	[(m s^−1^)^−1^ $d_{x,t}^{-1}$]

**Table 10 entropy-24-01493-t010:** Conservation Laws for the Volumetric-Velocitotemporal Formulation (based on the Velocivolumetric Fluid Density ζ(u,x,t)).

Conserved quantity	Density	Integral Equation			Differential Equation		
	φ(u,x,t) $=\zeta \;\overset{˘}{\underline{\varphi }}$	LHS = ${▾}_{u}\beta$ $={▾}_{u}\left(Д\overset{˘}{\beta }\right)$	= RHS	SI Units	LHS	= RHS	SI Units
Fluid mass	ζ	${▾}_{u}Д$ $={▾}_{u}\left[\;\left[\;\left[\zeta \right]\;\right]\;\right]$	$=\underset{\Omega (u,t)}{\;\int \int \int \;}\left({▾}_{u}\zeta +\nabla_{x}\cdot \left(\zeta \;{\tilde{\Gamma }}^{⊤}\right)\right)dV$	[kg (m s^−1^)^−3^ $d_{u,t}^{-1}$]	$L_{{\tilde{\Gamma }}^{⊤}∟u,t}^{\left(u,t\right)}\left(\zeta {\mathrm{vol}}_{x}^{3}\right)$	$=\left({▾}_{u}\zeta +\nabla_{x}\cdot \left(\zeta \;{\tilde{\Gamma }}^{⊤}\right)\right)dV$	[kg (m s^−1^)^−3^ $d_{u,t}^{-1}$]
Species mass	$\zeta {\overset{˘}{\underline{\chi }}}_{c}$ $=\zeta_{c}$	$\left.\begin{array}{c} {▾}_{u}\left(Д{\overset{˘}{\chi }}_{c}\right)={▾}_{u}\left({Д}_{c}\right)\\ ={▾}_{u}\left[\;\left[\;\left[\zeta {\overset{˘}{\underline{\chi }}}_{c}\right]\;\right]\;\right] \end{array}\right\}$	$=\underset{\Omega (u,t)}{\;\int \int \int \;}\left({▾}_{u}\left(\zeta {\overset{˘}{\underline{\chi }}}_{c}\right)+\nabla_{x}\cdot \left(\zeta {\overset{˘}{\underline{\chi }}}_{c}\;{\tilde{\Gamma }}^{⊤}\right)\right)dV$	[kg_c_ (m s^−1^)^−3^ $d_{u,t}^{-1}$]	$L_{{\tilde{\Gamma }}^{⊤}∟u,t}^{\left(u,t\right)}\left(\zeta {\overset{˘}{\underline{\chi }}}_{c}{\mathrm{vol}}_{x}^{3}\right)$	$=\left({▾}_{u}\left(\zeta {\overset{˘}{\underline{\chi }}}_{c}\right)+\nabla_{x}\cdot \left(\zeta {\overset{˘}{\underline{\chi }}}_{c}\;{\tilde{\Gamma }}^{⊤}\right)\right)dV$	[kg_c_ (m s^−1^)^−3^ $d_{u,t}^{-1}$]
Linear momentum	ζu	${▾}_{u}\left(Дu\right)$ $={▾}_{u}\left[\;\left[\;\left[\zeta u\right]\;\right]\;\right]$	$=\underset{\Omega (u,t)}{\;\int \int \int \;}\left({▾}_{u}\left(\zeta u\right)+\nabla_{x}\cdot \left(\zeta u\;{\tilde{\Gamma }}^{⊤}\right)\right)dV$	[(kg m s^−1^) (m s^−1^)^−3^ $d_{u,t}^{-1}$]	$L_{{\tilde{\Gamma }}^{⊤}∟u,t}^{\left(u,t\right)}\left(\zeta u{\mathrm{vol}}_{x}^{3}\right)$	$=\left({▾}_{u}\left(\zeta u\right)+\nabla_{x}\cdot \left(\zeta u\;{\tilde{\Gamma }}^{⊤}\right)\right)dV$	[(kg m s^−1^) (m s^−1^)^−3^ $d_{u,t}^{-1}$]
Angular momentum	$\zeta \left(\underline{\overset{˘}{r}}\times u\right)$	$\left.\begin{array}{c} {▾}_{u}\left(Д(\overset{˘}{r}\times u)\right)\\ ={▾}_{u}\left[\;\left[\;\left[\zeta \left(\underline{\overset{˘}{r}}\times u\right)\right]\;\right]\;\right] \end{array}\right\}$	$=\left\{\begin{array}{c} \underset{\Omega (u,t)}{\;\int \int \int \;}({▾}_{u}\left(\zeta \left(\underline{\overset{˘}{r}}\times u\right)\right)\\ +\nabla_{x}\cdot \left(\zeta \left(\underline{\overset{˘}{r}}\times u\right)\;{\tilde{\Gamma }}^{⊤}\right))dV \end{array}\right.$	[(kg m^2^ s^−1^) (m s^−1^)^−3^ $d_{u,t}^{-1}$]	$\left.\begin{array}{c} L_{{\tilde{\Gamma }}^{⊤}∟u,t}^{\left(u,t\right)}(\zeta \left(\underline{\overset{˘}{r}}\times u\right)\\ {\mathrm{vol}}_{x}^{3}) \end{array}\right\}$	$=\left\{\begin{array}{c} ({▾}_{u}\left(\zeta \left(\underline{\overset{˘}{r}}\times u\right)\right)\\ +\nabla_{x}\cdot \left(\zeta \left(\underline{\overset{˘}{r}}\times u\right)\;{\tilde{\Gamma }}^{⊤}\right))dV \end{array}\right.$	[(kg m^2^ s^−1^) (m s^−1^)^−3^ $d_{u,t}^{-1}$]
Energy	$\zeta \overset{˘}{\underline{e}}$	${▾}_{u}\left(Д\overset{˘}{e}\right)$ $={▾}_{u}\left[\;\left[\;\left[\zeta \overset{˘}{\underline{e}}\right]\;\right]\;\right]$	$=\underset{\Omega (u,t)}{\;\int \int \int \;}\left({▾}_{u}\left(\zeta \overset{˘}{\underline{e}}\right)+\nabla_{x}\cdot \left(\zeta \overset{˘}{\underline{e}}\;{\tilde{\Gamma }}^{⊤}\right)\right)dV$	[J (m s^−1^)^−3^ $d_{u,t}^{-1}$]	$L_{{\tilde{\Gamma }}^{⊤}∟u,t}^{\left(u,t\right)}\left(\zeta \overset{˘}{\underline{e}}{\mathrm{vol}}_{x}^{3}\right)$	$=\left({▾}_{u}\left(\zeta \overset{˘}{\underline{e}}\right)+\nabla_{x}\cdot \left(\zeta \overset{˘}{\underline{e}}\;{\tilde{\Gamma }}^{⊤}\right)\right)dV$	[J (m s^−1^)^−3^ $d_{u,t}^{-1}$]
Charge (in solution)	$\zeta \overset{˘}{\underline{z}}$	${▾}_{u}\left(Д\overset{˘}{z}\right)$ $={▾}_{u}\left[\;\left[\;\left[\zeta \overset{˘}{\underline{z}}\right]\;\right]\;\right]$	$=\underset{\Omega (u,t)}{\;\int \int \int \;}\left({▾}_{u}\left(\zeta \overset{˘}{\underline{z}}\right)+\nabla_{x}\cdot \left(\zeta \overset{˘}{\underline{z}}\;{\tilde{\Gamma }}^{⊤}\right)\right)dV$	[C (m s^−1^)^−3^ $d_{u,t}^{-1}$]	$L_{{\tilde{\Gamma }}^{⊤}∟u,t}^{\left(u,t\right)}\left(\zeta \overset{˘}{\underline{z}}{\mathrm{vol}}_{x}^{3}\right)$	$=\left({▾}_{u}\left(\zeta \overset{˘}{\underline{z}}\right)+\nabla_{x}\cdot \left(\zeta \overset{˘}{\underline{z}}\;{\tilde{\Gamma }}^{⊤}\right)\right)dV$	[C (m s^−1^)^−3^ $d_{u,t}^{-1}$]
Entropy	$\zeta \overset{˘}{\underline{s}}$	${▾}_{u}\left(Д\overset{˘}{s}\right)$ $={▾}_{u}\left[\;\left[\;\left[\zeta \overset{˘}{\underline{s}}\right]\;\right]\;\right]$	$=\underset{\Omega (u,t)}{\;\int \int \int \;}\left({▾}_{u}\left(\zeta \overset{˘}{\underline{s}}\right)+\nabla_{x}\cdot \left(\zeta \overset{˘}{\underline{s}}\;{\tilde{\Gamma }}^{⊤}\right)\right)dV$	[J K^−1^ (m s^−1^)^−3^ $d_{u,t}^{-1}$]	$L_{{\tilde{\Gamma }}^{⊤}∟u,t}^{\left(u,t\right)}\left(\zeta \overset{˘}{\underline{s}}{\mathrm{vol}}_{x}^{3}\right)$	$=({▾}_{u}\left(\zeta \overset{˘}{\underline{s}}\right)+\nabla_{x}\cdot \left(\zeta \overset{˘}{\underline{s}}\;{\tilde{\Gamma }}^{⊤}\right))dV$	[J K^−1^ (m s^−1^)^−3^$d_{u,t}^{-1}$]
Probability	p(u,x|t) $=p_{u,x|t}$	$\left.\begin{array}{c} {▾}_{u}p_{u|t}\\ ={▾}_{u}\left[\;\left[\;\left[p_{u,x|t}\right]\;\right]\;\right] \end{array}\right\}$	$=\underset{\Omega (u,t)}{\;\int \int \int \;}\left({▾}_{u}p_{u,x|t}+\nabla_{x}\cdot \left(p_{u,x|t}\;{\tilde{\Gamma }}^{⊤}\right)\right)dV$	[(m s^−1^)^−3^ $d_{u,t}^{-1}$]	$\left.\begin{array}{c} L_{{\tilde{\Gamma }}^{⊤}∟u,t}^{\left(u,t\right)}\\ \left(p_{u,x|t}{\mathrm{vol}}_{x}^{3}\right) \end{array}\right\}$	$=\left({▾}_{u}p_{u,x|t}+\nabla_{x}\cdot \left(p_{u,x|t}\;{\tilde{\Gamma }}^{⊤}\right)\right)dV$	[(m s^−1^)^−3^ $d_{u,t}^{-1}$]
	p(x|u,t) $=p_{x|u,t}$	$0={▾}_{u}\left[\;\left[\;\left[p_{x|u,t}\right]\;\right]\;\right]$	$=\underset{\Omega (u,t)}{\;\int \int \int \;}\left({▾}_{u}p_{x|u,t}+\nabla_{x}\cdot \left(p_{x|u,t}\;{\tilde{\Gamma }}^{⊤}\right)\right)dV$	[$d_{u,t}^{-1}$]	0	$={▾}_{u}p_{x|u,t}+\nabla_{x}\cdot \left(p_{x|u,t}\;{\tilde{\Gamma }}^{⊤}\right)$	[m^−3^ $d_{u,t}^{-1}$]

**Table 11 entropy-24-01493-t011:** Conservation Laws for the Velocimetric-Temporal Formulation (based on the Velocimetric Fluid Density η(u,x,t)).

Conserved Quantity	Density	Integral Equation			Differential Equation		
	θ(u,x,t) $=\eta \overset{˘}{\underline{\theta }}$	LHS = $\frac{dQ}{dt}$ = $\frac{DQ}{Dt}$	= RHS	SI Units	LHS	= RHS	SI Units
Fluid mass	η	$0=\dot{M}$ $=\frac{d\langle \;\langle \;\langle \eta \rangle \;\rangle \;\rangle }{dt}$	$=\underset{D(x,t)}{\;\int \int \int \;}\left[\frac{\partial \eta }{\partial t}+\nabla_{u}\cdot \left(\eta \dot{u}\right)\right]dU$	[kg s^−1^]	0	$=\frac{\partial \eta }{\partial t}+\nabla_{u}\cdot \left(\eta \dot{u}\right)$	[kg s^−1^ (m s^−1^)^−3^]
Species mass	$\eta {\overset{˘}{\underline{\chi }}}_{c}$ = $\eta_{c}$	$\left({\dot{M}}_{c}\right)=\frac{d\langle \;\langle \;\langle \eta {\overset{˘}{\underline{\chi }}}_{c}\rangle \;\rangle \;\rangle }{dt}$	$=\underset{D(x,t)}{\;\int \int \int \;}\left[\frac{\partial \eta {\overset{˘}{\underline{\chi }}}_{c}}{\partial t}+\nabla_{u}\cdot \left(\eta {\overset{˘}{\underline{\chi }}}_{c}\dot{u}\right)\right]dU$	[kg_c_ s^−1^]	$\eta \frac{d{\overset{˘}{\underline{\chi }}}_{c}}{dt}$	$=\frac{\partial \eta {\overset{˘}{\underline{\chi }}}_{c}}{\partial t}+\nabla_{u}\cdot \left(\eta {\overset{˘}{\underline{\chi }}}_{c}\dot{u}\right)$	[kg_c_ s^−1^ (m s^−1^)^−3^]
Linear momentum	ηu	$\sum F=\frac{d\langle \;\langle \;\langle \eta u\rangle \;\rangle \;\rangle }{dt}$	$=\underset{D(x,t)}{\;\int \int \int \;}\left[\frac{\partial \eta u}{\partial t}+\nabla_{u}\cdot \left(\eta u\dot{u}\right)\right]dU$	[(kg m s^−1^) s^−1^ = N]	$\eta \frac{du}{dt}$	$=\frac{\partial \eta u}{\partial t}+\nabla_{u}\cdot \left(\eta u\dot{u}\right)$	[N (m s^−1^)^−3^]
Angular momentum	$\eta (\overset{˘}{\underline{r}}\times u)$	$\sum T=\frac{d\langle \;\langle \;\langle \eta \left(\overset{˘}{\underline{r}}\times u\right)\rangle \;\rangle \;\rangle }{dt}$	$=\underset{D(x,t)}{\;\int \int \int \;}[\frac{\partial \eta (\overset{˘}{\underline{r}}\times u)}{\partial t}$ $+\nabla_{u}\cdot \left(\eta \left(\overset{˘}{\underline{r}}\times u\right)\dot{u}\right)]dU$	[(kg m${}^{2}$ s^−1^) s^−1^ = N m]	$\eta \frac{d(\overset{˘}{\underline{r}}\times u)}{dt}$	$=\frac{\partial \eta (\overset{˘}{\underline{r}}\times u)}{\partial t}$ $+\nabla_{u}\cdot \left(\eta \left(\overset{˘}{\underline{r}}\times u\right)\dot{u}\right)$	[(N m) (m s^−1^)^−3^]
Energy	$\eta \overset{˘}{\underline{e}}$	$\left.\begin{array}{c} \frac{DE}{Dt}=\frac{d\langle \;\langle \;\langle \eta \overset{˘}{\underline{e}}\rangle \;\rangle \;\rangle }{dt}\\ =({\dot{Q}}_{in}+{\dot{W}}_{in}) \end{array}\right\}$	$=\underset{D(x,t)}{\;\int \int \int \;}\left[\frac{\partial \eta \overset{˘}{\underline{e}}}{\partial t}+\nabla_{u}\cdot \left(\eta \overset{˘}{\underline{e}}\dot{u}\right)\right]dU$	[J s^−1^ = W]	$\eta \frac{d\overset{˘}{\underline{e}}}{dt}$	$=\frac{\partial \eta \overset{˘}{\underline{e}}}{\partial t}+\nabla_{u}\cdot \left(\eta \overset{˘}{e}\dot{u}\right)$	[J s^−1^ (m s^−1^)^−3^ = W (m s^−1^)^−3^]
Charge (in solution)	$\eta \overset{˘}{\underline{z}}$	$\left.\begin{array}{c} \frac{DZ}{Dt}=\frac{d\langle \;\langle \;\langle \eta \overset{˘}{\underline{z}}\rangle \;\rangle \;\rangle }{dt}\\ =I+\left(\sum_{c}z_{c}{\dot{M}}_{c}\right) \end{array}\right\}$	$=\underset{D(x,t)}{\;\int \int \int \;}\left[\frac{\partial \eta \overset{˘}{\underline{z}}}{\partial t}+\nabla_{u}\cdot \left(\eta \overset{˘}{\underline{z}}\dot{u}\right)\right]dU$	[C s^−1^ = A]	$\eta \frac{d\overset{˘}{\underline{z}}}{dt}$	$=\frac{\partial \eta \overset{˘}{\underline{z}}}{\partial t}+\nabla_{u}\cdot \left(\eta \overset{˘}{\underline{z}}\dot{u}\right)$	[C s^−1^ (m s^−1^)^−3^ = A (m s^−1^)^−3^]
Entropy	$\eta \overset{˘}{\underline{s}}$	$\left.\begin{array}{c} \frac{DS}{Dt}=\frac{d\langle \;\langle \;\langle \eta \overset{˘}{\underline{s}}\rangle \;\rangle \;\rangle }{dt}\\ =\dot{\sigma }+{\dot{S}}^{nf} \end{array}\right\}$	$=\underset{D(x,t)}{\;\int \int \int \;}\left[\frac{\partial \eta \overset{˘}{\underline{s}}}{\partial t}+\nabla_{u}\cdot \left(\eta \overset{˘}{\underline{s}}\dot{u}\right)\right]dU$	[J K^−1^ s^−1^]	$\eta \frac{d\overset{˘}{\underline{s}}}{dt}$	$=\frac{\partial \eta \overset{˘}{\underline{s}}}{\partial t}+\nabla_{u}\cdot \left(\eta \overset{˘}{\underline{s}}\dot{u}\right)$	[J K^−1^ s^−1^ (m s^−1^)^−3^]
Probability	p(u|x,t) $=p_{u|x,t}$	$0=\frac{d\langle \;\langle \;\langle p_{u|x,t}\rangle \;\rangle \;\rangle }{dt}$	$=\underset{D(x,t)}{\;\int \int \int \;}\left[\frac{\partial p_{u|x,t}}{\partial t}+\nabla_{u}\cdot \left(p_{u|x,t}\dot{u}\right)\right]dU$	[s^−1^]	0	$=\frac{\partial p_{u|x,t}}{\partial t}+\nabla_{u}\cdot \left(p_{u|x,t}\dot{u}\right)$	[(m s^−1^)^−3^ s^−1^]

**Table 12 entropy-24-01493-t012:** Conservation Laws for the Volumetric-Temporal Formulation (based on the Volumetric Fluid Density ξ(u,x,t)).

Conserved quantity	Density	Integral Equation			Differential Equation		
	ϵ(u,x,t) $=\xi \overset{˘}{\underline{\epsilon }}$	LHS = $\frac{dQ}{dt}$ = $\frac{DQ}{Dt}$	= RHS	SI Units	LHS	= RHS	SI Units
Fluid mass	ξ	$0=\dot{M}$ $=\frac{d[\;[\;[\xi ]\;]\;]}{dt}$	$=\underset{\Omega (u,t)}{\;\int \int \int \;}\left[\frac{\partial \xi }{\partial t}+\nabla_{x}\cdot \left(\xi u\right)\right]dV$	[kg s^−1^]	0	$=\frac{\partial \xi }{\partial t}+\nabla_{x}\cdot \left(\xi u\right)$	[kg s^−1^ m^−3^]
Species mass	$\xi {\overset{˘}{\underline{\chi }}}_{c}$ $=\xi_{c}$	$\left({\dot{M}}_{c}\right)=\frac{d[\;\left[\;\left[\xi {\overset{˘}{\underline{\chi }}}_{c}\right]\;\right]\;]}{dt}$	$=\underset{\Omega (u,t)}{\;\int \int \int \;}\left[\frac{\partial \xi {\overset{˘}{\underline{\chi }}}_{c}}{\partial t}+\nabla_{x}\cdot \left(\xi {\overset{˘}{\underline{\chi }}}_{c}u\right)\right]dV$	[kg_c_ s^−1^]	$\xi \frac{D{\overset{˘}{\underline{\chi }}}_{c}}{Dt}$	$=\frac{\partial \xi {\overset{˘}{\underline{\chi }}}_{c}}{\partial t}$ $+\nabla_{x}\cdot \left(\xi {\overset{˘}{\underline{\chi }}}_{c}u\right)$	[kg_c_ s^−1^ m^−3^]
Linear momentum	ξu	$\sum F=\frac{d[\;[\;[\xi u]\;]\;]}{dt}$	$=\underset{\Omega (u,t)}{\;\int \int \int \;}\left[\frac{\partial \xi u}{\partial t}+\nabla_{x}\cdot \left(\xi uu\right)\right]dV$	[(kg m s^−1^) s^−1^ = N]	$\xi \frac{Du}{Dt}$	$=\frac{\partial \xi u}{\partial t}+\nabla_{x}\cdot \left(\xi uu\right)$	[N m^−3^]
Angular momentum	$\xi (\overset{˘}{\underline{r}}\times u)$	$\sum T=\frac{d[\;\left[\;\left[\xi \left(\overset{˘}{\underline{r}}\times u\right)\right]\;\right]\;]}{dt}$	$=\underset{\Omega (u,t)}{\;\int \int \int \;}[\frac{\partial \xi (\overset{˘}{\underline{r}}\times u)}{\partial t}$ $+\nabla_{x}\cdot \left(\xi \left(\overset{˘}{\underline{r}}\times u\right)u\right)]dV$	[(kg m${}^{2}$ s^−1^) s^−1^ = N m]	$\xi \frac{D(\overset{˘}{\underline{r}}\times u)}{Dt}$	$=\frac{\partial \xi (\overset{˘}{\underline{r}}\times u)}{\partial t}$ $+\nabla_{x}\cdot \left(\xi \left(\overset{˘}{\underline{r}}\times u\right)u\right)$	[N m^−2^ = (N m) m^−3^]
Energy	$\xi \overset{˘}{\underline{e}}$	$\left.\begin{array}{c} \frac{DE}{Dt}=\frac{d[\;\left[\;\left[\xi \overset{˘}{\underline{e}}\right]\;\right]\;]}{dt}\\ =({\dot{Q}}_{in}+{\dot{W}}_{in}) \end{array}\right\}$	$=\underset{\Omega (u,t)}{\;\int \int \int \;}\left[\frac{\partial \xi \overset{˘}{\underline{e}}}{\partial t}+\nabla_{x}\cdot \left(\xi \overset{˘}{\underline{e}}u\right)\right]dV$	[J s^−1^ = W]	$\xi \frac{D\overset{˘}{\underline{e}}}{Dt}$	$=\frac{\partial \xi \overset{˘}{\underline{e}}}{\partial t}+\nabla_{x}\cdot \left(\xi \overset{˘}{\underline{e}}u\right)$	[J s^−1^ m^−3^ = W m^−3^]
Charge (in solution)	$\xi \overset{˘}{\underline{z}}$	$\left.\begin{array}{c} \frac{DZ}{Dt}=\frac{d[\;\left[\;\left[\xi \overset{˘}{\underline{z}}\right]\;\right]\;]}{dt}\\ =I+\left(\sum_{c}z_{c}{\dot{M}}_{c}\right) \end{array}\right\}$	$=\underset{\Omega (u,t)}{\;\int \int \int \;}\left[\frac{\partial \xi \overset{˘}{\underline{z}}}{\partial t}+\nabla_{x}\cdot \left(\xi \overset{˘}{\underline{z}}u\right)\right]dV$	[C s^−1^ = A]	$\xi \frac{D\overset{˘}{\underline{z}}}{Dt}$	$=\frac{\partial \xi \overset{˘}{\underline{z}}}{\partial t}+\nabla_{x}\cdot \left(\xi \overset{˘}{\underline{z}}u\right)$	[C s^−1^ m^−3^ = A m^−3^]
Entropy	$\xi \overset{˘}{\underline{s}}$	$\left.\begin{array}{c} \frac{DS}{Dt}=\frac{d[\;\left[\;\left[\xi \overset{˘}{\underline{s}}\right]\;\right]\;]}{dt}\\ =\dot{\sigma }+{\dot{S}}^{nf} \end{array}\right\}$	$=\underset{\Omega (u,t)}{\;\int \int \int \;}\left[\frac{\partial \xi \overset{˘}{\underline{s}}}{\partial t}+\nabla_{x}\cdot \left(\xi \overset{˘}{\underline{s}}u\right)\right]dV$	[J K^−1^ s^−1^]	$\xi \frac{D\overset{˘}{\underline{s}}}{Dt}$	$=\frac{\partial \xi \overset{˘}{\underline{s}}}{\partial t}+\nabla_{x}\cdot \left(\xi \overset{˘}{\underline{s}}u\right)$	[J K^−1^ s^−1^ m^−3^]
Probability	p(x|u,t) $=p_{x|u,t}$	$0=\frac{d[\;\left[\;\left[p_{x|u,t}\right]\;\right]\;]}{dt}$	$=\underset{\Omega (u,t)}{\;\int \int \int \;}\left[\frac{\partial p_{x|u,t}}{\partial t}+\nabla_{x}\cdot \left(p_{x|u,t}u\right)\right]dV$	[s^−1^]	0	$=\frac{\partial p_{x|u,t}}{\partial t}+\nabla_{x}\cdot \left(p_{x|u,t}u\right)$	[s^−1^ m^−3^]

## Data Availability

No data were generated in this study.

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
