# Peer review of "A Hierarchy of Probability, Fluid and Generalized Densities for the Eulerian Velocivolumetric Description of Fluid Flow, for New Families of Conservation Laws"

_entropy, 2022, doi:10.3390/e24101493_

Round 1

Reviewer 2 Report

In this manuscript, the authors present a very interesting work on the generalization of the Reynolds transport theorem.

The presented work is very impressive and I definitively think that it represents a great contribution for the scientific community. It is based on the results of two cited paper of the same authors, but with a more extensive generalization. The starting point are conservation laws of probabilities. Applying elementary observations to probability distributions, generalizations of conservation laws of a variety different quantities are deduced. Old conservation theorems are recovered in the developed framework, but also completely new families of conservation laws.

The article is well written and clearly presented as well as the very complete list of applications laws. I recommend this article for publication.

I have just few suggestions that should not affect the publication decision.

1)     Figure 4: I suggest to put figure 4a) on top of 4b) and 4c)

2)    About possible outlooks on extension of the present work, I would discuss the possibility to extend the theorems in the case of special relativity. In the present manuscript, the authors considered non-infinite velocities.  In the relativistic case velocity would be bounded by the velocity of light automatically. Moreover, new rules could be appeared because of the relation between mass and energy. Such a relativistic scenario would be interesting for relativistic plasmas dynamics.
